# Gaussian-Smoothed Sliced Probability Divergences

**Mokhtar Z. Alaya**  *alayaelm@utc.fr*
*Université de Technologie de Compiègne,*
*LMAC (Laboratoire de Mathématiques Appliquées de Compiègne), CS 60 319 - 60 203 Compiègne Cedex*

**Alain Rakotomamonjy**  *a.rakotomamonjy@criteo.com*
*Criteo AI Lab, Paris, France,*

**Maxime Berar**  *maxime.berar@univ-rouen.fr*
*Univ Rouen Normandie, INSA Rouen Normandie, Universite Le Havre Normandie*
*Normandie Univ, LITIS UR4108, Rouen, France*

**Gilles Gasso**  *gilles.gasso@insa-rouen.fr*
*INSA Rouen Normandie, Univ Rouen Normandie, Universite Le Havre Normandie,*
*Normandie Univ, LITIS UR4108, Rouen, France*

**Reviewed on OpenReview:** *https://openreview.net/forum?id=weuALLWUV2*

## Abstract

Gaussian smoothed sliced Wasserstein distance has been recently introduced for comparing probability distributions, while preserving privacy on the data. It has been shown that it provides performances similar to its non-smoothed (non-private) counterpart. However, the computational and statistical properties of such a metric have not yet been well-established. This work investigates the theoretical properties of this distance as well as those of generalized versions denoted as Gaussian-smoothed sliced divergences $\mathrm{G}_\sigma\mathrm{SD}_p$. We first show that smoothing and slicing preserve the metric property and the weak topology. To study the sample complexity of such divergences, we then introduce $\hat{\hat{\mu}}_n$ the double empirical distribution for the smoothed-projected $\mu$. The distribution $\hat{\hat{\mu}}_n$ is a result of a double sampling process: one from sampling according to the origin distribution $\mu$ and the second according to the convolution of the projection of $\mu$ on the unit sphere and the Gaussian smoothing. We particularly focus on the Gaussian smoothed sliced Wasserstein distance $\mathrm{G}_\sigma\mathrm{SW}_p$ and prove that it converges with a rate $O(n^{-1/2p})$. We also derive other properties, including continuity, of different divergences with respect to the smoothing parameter. We support our theoretical findings with empirical studies in the context of privacy-preserving domain adaptation.

## 1 Introduction

Divergences for comparing two distributions have been shown to be important for achieving good performance in the contexts of generative modeling (Arjovsky et al., 2017; Salimans et al., 2018), domain adaptation (Long et al., 2015; Courty et al., 2016; Lee et al., 2019), and in computer vision (Bonneel et al., 2011; Solomon et al., 2015) among many more applications (Kolouri et al., 2017; Peyré & Cuturi, 2019; Nguyen et al., 2023). Examples of divergences that have proved useful for these tasks are the Maximum Mean Discrepancy (Gretton et al., 2012; Long et al., 2015; Sutherland et al., 2017), the Wasserstein distance (Monge, 1781; Kantorovich, 1942; Villani, 2009) or its variant the sliced Wasserstein distance (SW) (Kolouri et al., 2016; Bonneel & Coeurjolly, 2019; Kolouri et al., 2019b; Nguyen et al., 2021; 2022; 2024).

The SW distance has the advantage of being computationally efficient, since it uses a closed-form solution for distributions with support on $\mathbb{R}$, by computing the expectation of one-dimensional (1D) random projections of distributions in $\mathbb{R}^d$. Owing to this efficiency and the resulting scalability, this distance has been successfully applied in several applications ranging from generative models to domain adaptation (Kolouri et al., 2019a;

Deshpande et al., 2019; Wu et al., 2019; Lee et al., 2019) and its statistical properties have been well-studied in Nadjahi et al. (2020).

Recently, Gaussian smoothed variants of the Wasserstein distance and the sliced Wasserstein distance have been introduced respectively in (Nietert et al., 2021) and in Rakotomamonjy & Ralaivola (2021). One main motivation behind these variants is to provide a privacy guarantee for the distribution comparison task as Gaussian smoothing is known to be a mechanism for achieving differential privacy (Dwork et al., 2014). While the properties of the Gaussian smoothed Wasserstein distance have been extensively studied by Nietert et al. (2021), the properties of the Gaussian smoothed sliced Wasserstein distance have not been fully investigated yet although they are known to be more computationally efficient.

In this work, we focus on the slicing of Gaussian-smoothed measure discrepancies by providing theoretical properties of more general divergences induced by some base distances or divergences for distributions defined in $\mathbb{R}^d$. These base distances/divergences encompass Wasserstein, maximum mean discrepancy, Sinkhorn divergence. As for a main contribution, we first establish the topological properties of these divergences in term of a metrization of the weak topology and a semi-lower continuous property. Then we focus on the sample complexity of such divergences by introducing the *double empirical distribution* $\hat{\hat{\mu}}_n$ for the smoothed-projected origin distribution $\mu$. The new empirical distribution is a result of double sampling process: one from sampling according to the origin distribution and the second according to the convolution of the projection of $\mu$ on the unit sphere and the Gaussian smoothing. The introducing of $\hat{\hat{\mu}}_n$ is inspired from the implementation part: we sample $X_1, \ldots, X_n$ from the raw distribution $\mu$ to define $\hat{\mu}_n$ then project it on the unit sphere and smooth this projection with a Gaussian distribution. This smoothing is a continuous measure that needs to be sampled. For that reason, we add a double sampling and then provide $\hat{\hat{\mu}}_n$. We particularly focus on the Gaussian smoothed sliced Wasserstein distance.

Given the importance of the noise level in the privacy/utility trade-off achieved by the divergence, we investigate an order relation and a continuity result with respect to the noise level. These properties are of high impact as it supports a computationally cheap warm-start/fine-tuning procedure when looking for a privacy/utility compromise of the divergence. Our theoretical study is backed by some numerical experiments on toy problems and on domain adaptation illustrating how owing to the topology induced by our metric and its continuity, differential privacy comes almost for free (without loss of performance) and multiple models with different level of privacy can be cheaply computed.

**Comparison with previous works.** Here we highlight the position of this work compared to the most linked previous ones, in particular Nadjahi et al. (2020) and Rakotomamonjy & Ralaivola (2021). The work of Nadjahi et al. (2020) is focused on sliced Wasserstein distance and its statistical properties, however our work is based on the properties of the Gaussian smoothed with general divergences (e.g. Wasserstein, MMD, Sinkhorn divergence). We argue that the properties cannot be directly derived from (Nadjahi et al., 2020), especially the sample complexity result. In Rakotomamonjy & Ralaivola (2021), the authors investigated the smoothed Wasserstein distance and their theoretical finding was principally on proving the metric property, whereas we further investigate sample and projection complexities and the continuity properties w.r.t. the smoothing noise level. We emphasize that the novelty of the present paper consists in the theoretical properties derived from the definition of the empirical measure $\hat{\hat{\mu}}_n$. The smoothing of the raw measures, from a theoretical point view, is a continuous measure (see Lemma 3.5) that needs to be sampled. This entails to define the second sampling step and construct $\hat{\hat{\mu}}_n$, an empirical version for the smoothing projection of $\mu$. To the best of our knowledge, this work is the first introducing the double randomness in the case of smoothing optimal transport discrepancies. Recent works (Goldfeld et al., 2020; Nietert et al., 2021) addressed the smoothing Wasserstein an their theoretical results relied only on $\hat{\mu}_n$.

**Layout of the paper.** The paper is organized as follows: after introducing the notation and some background in Section 2, we detail the topological properties of Gaussian-smoothed sliced divergence in Section 3.1 while the double sampling process and its statistical properties are established in Section 3.2. The noise analyses are provided in Section 3.3. Experimental analyses for supporting the theory and showcasing the relevance of our divergences in domain adaptation are depicted in Section 4. Discussions on the perspectives and limitations are in Section 5. All the proofs of the theoretical results and some additional experiments are postponed to the appendices in the supplementary.

## 2  Preliminaries

For the reader's convenience, we provide a brief summary of standard notations and definitions used throughout the paper.

**Notation.**  For $d \in \mathbb{N}^*$, let $\mathcal{P}(\mathbb{R}^d)$ be the set of Borel probability measures on $\mathbb{R}^d$ and $\mathcal{P}_p(\mathbb{R}^d) \subset \mathcal{P}(\mathbb{R}^d)$, those with finite moment of order $p$, i.e., $\mathcal{P}_p(\mathbb{R}^d) \triangleq \{\mu \in \mathcal{P} : \int \|x\|^p d\mu(x) < \infty\}$, where $\|\cdot\|$ is the Euclidean norm. We denote $M_p(\mu) = \int_x \|x\|^p \mathrm{d}\mu(x)$. For two probability distributions $\mu$ and $\nu$, we denote their convolution as $\mu * \nu \in \mathcal{P}(\mathbb{R}^d)$, namely $(\mu * \nu)(A) = \int_x \int_y \mathbf{1}_A(x+y)\mathrm{d}\mu(x)\mathrm{d}\nu(y)$, where $\mathbf{1}_A(\cdot)$ is the indicator function over $A$. Given two independent random variables $X \sim \mu$ and $Y \sim \nu$, we remind that $X + Y \sim \mu * \nu$. The $d$-dimensional unit-sphere is noted as $\mathbb{S}^{d-1} \triangleq \{\theta \in \mathbb{R}^d : \|\theta\| = 1\}$. We denote by $u_d$ the uniform distribution on $\mathbb{S}^{d-1}$ and we use $\delta(\cdot)$ to denote the Kronecker delta function. We note as $\mathbf{E}_\mu f$ the expectation of the function $f$ with respect to $\mu$.

Let $\Gamma : \mathbb{R} \to \mathbb{R}$ be the Gamma function expressed as $\Gamma(v) = \int_0^\infty t^{v-1}e^{-t}dt$ for $v > 0$. For $k \in \mathbb{N}$, $(\cdot)_k$ denoted the Pochhammer symbol, also known in the literature as a rising factorial, namely $(\alpha)_0 = 1, (\alpha)_1 = \alpha$, and $(\alpha)_k = \frac{\Gamma(\alpha+k)}{\Gamma(k)} = \alpha(\alpha+1)\cdots(\alpha+k-1)$, for $k \geq 1$. We denote by $_1F_1(\alpha, \gamma; z)$ the Kummers confluent hypergeometric function (Olver, 2010) and defined by $_1F_1(\alpha, \gamma; z) = \sum_{k=0}^\infty \frac{(\alpha)_k}{(\gamma)_k} \frac{z^k}{k!}$.

**Sliced Wasserstein distance.**  We remind in this paragraph several measures of similarity between two distributions. The Wasserstein distance of order $p \in [1, \infty)$ between two measures in $\mathcal{P}_p(\mathbb{R}^d)$ is given by the relaxation of the optimal transport problem, and it is defined as

$$\mathrm{W}_p(\mu, \nu) = \left( \inf_{\gamma \in \Pi(\mu,\nu)} \int_{\mathbb{R}^d \times \mathbb{R}^d} \|x - x'\|^p \gamma(x, x')\mathrm{d}x\mathrm{d}x' \right)^{1/p}$$

where $\Pi(\mu, \nu) \triangleq \{\gamma \in \mathcal{P}(\mathbb{R}^d \times \mathbb{R}^d) | \pi_{1\#}\gamma = \mu, \pi_{2\#}\gamma = \nu\}$ and $\pi_1, \pi_2$ are the marginal projectors of $\gamma$ on each of its coordinates. When $d = 1$, the Wasserstein distance can be calculated in closed-form owing to the cumulative distributions of $\mu$ and $\nu$ (Rachev & Rüschendorf, 1998). In practice for empirical distributions, the closed-form solution requires only the sorting of the samples, which makes it very efficient. Because of this efficiency, efforts have been devoted to derive a metric for high-dimensional distributions based on 1D Wasserstein distance. The main idea is to project high-dimensional probability distributions onto a random one-dimensional space and then to compute the Wasserstein distance. This operation can be theoretically formalized through the use of the Radon transform, leading to the so-called sliced Wasserstein distance (Kolouri et al., 2016; Bonneel & Coeurjolly, 2019; Kolouri et al., 2019b; Nguyen et al., 2021).

**Definition 2.1.** For any $p \in [1, \infty)$ and two measures $\mu, \nu \in \mathcal{P}_p(\mathbb{R}^d)$, the sliced Wasserstein distance (SW) reads as

$$\mathrm{SW}_p(\mu, \nu) \triangleq \left( \int_{\mathbb{S}^{d-1}} \mathrm{W}_p^p(\mathcal{R}_\mathbf{u}\mu, \mathcal{R}_\mathbf{u}\nu) u_d(\mathbf{u})\mathrm{d}\mathbf{u} \right)^{1/p}.$$

where $\mathcal{R}_\mathbf{u}$ is the Radon transform of a probability distribution, namely $\mathcal{R}_\mathbf{u}\mu(\cdot) = \int_{\mathbb{R}^d} \mu(\mathbf{s})\delta(\cdot - \mathbf{s}^\top \mathbf{u})d\mathbf{s}$. In practice, the integral is approximated through a Monte-Carlo simulation leading to a sum of 1D Wasserstein distances over a fixed number of random directions $\mathbf{u}$.

**Gaussian-smoothed sliced Wasserstein distance.**  Based on this definition of SW, replacing the Radon projected measures with their Gaussian-smoothed counterpart leads to the following definition:

**Definition 2.2.** The $\sigma$-Gaussian-smoothed $p$-Sliced Wasserstein distance between probability distributions $\mu$ and $\nu$ in $\mathcal{P}_p(\mathbb{R}^d)$ writes as

$$\mathrm{G}_\sigma\mathrm{SW}_p(\mu, \nu) \triangleq \left( \int_{\mathbb{S}^{d-1}} \mathrm{W}_p^p(\mathcal{R}_\mathbf{u}\mu * \mathcal{N}_\sigma, \mathcal{R}_\mathbf{u}\nu * \mathcal{N}_\sigma) u_d(\mathbf{u})\mathrm{d}\mathbf{u} \right)^{1/p},$$

where $\mathcal{N}_\sigma = \mathcal{N}(0, \sigma^2)$ is the zero-mean $\sigma^2$-variance Gaussian measure. It is important to note here that the smoothing (convolution) operation occurs after projection onto the one-dimensional space. Hence, assuming $X \sim \mu$, $Y \sim \nu$, for a given direction $\mathbf{u}$, we compute in the integral the one-dimensional Wasserstein distance between the probability laws of $\mathbf{u}^\top X + Z$ and $\mathbf{u}^\top Y + Z'$ where $Z, Z' \sim \mathcal{N}_\sigma$ are independent random variables. The metric properties of $\mathrm{G}_\sigma \mathrm{SW}_p$ for $p \geq 1$ have been discussed in a recent work (Rakotomamonjy & Ralaivola, 2021). This latter work has also shown, in the context of differential privacy, the importance of convolving the Radon projected distribution with a Gaussian instead of computing the SW distance of the original distribution smoothed with a $d$-dimensional Gaussian $\mu * \mathcal{N}_{\sigma \mathbf{I}_d}$, where $\mathbf{I}_d$ denotes the $d \times d$ identity matrix.

**Gaussian-smoothed sliced divergence.** The idea of slicing high-dimensional distributions before feeding them to a divergence between probability distributions can be extended to distances other than the Wasserstein distance. These sliced divergences have been studied by Nadjahi et al. (2020). Similarly, we can define a Gaussian-smoothed sliced divergence, given a divergence $\mathrm{D}_{\mathbb{R}^d} : \mathcal{P}_p(\mathbb{R}^d) \times \mathcal{P}_p(\mathbb{R}^d) \to \mathbb{R}^+$ for $d \geq 1$ as:

**Definition 2.3.** The $\sigma$-Gaussian-smoothed $p$-Sliced Divergence between probability distributions $\mu$ and $\nu$ in $\mathcal{P}_p(\mathbb{R}^d)$ associated to the *base divergence* $\mathrm{D} \triangleq \mathrm{D}_{\mathbb{R}}$, $p \geq 1$ is

$$\mathrm{G}_\sigma \mathrm{SD}_p(\mu, \nu) \triangleq \left( \int_{\mathbb{S}^{d-1}} \mathrm{D}^p(\mathcal{R}_\mathbf{u}\mu * \mathcal{N}_\sigma, \mathcal{R}_\mathbf{u}\nu * \mathcal{N}_\sigma) u_d(\mathbf{u}) \mathrm{d}\mathbf{u} \right)^{1/p}.$$

Typical relevant divergences are the maximum mean discrepancy (MMD) (Gretton et al., 2012) or the Sinkhorn divergence (Genevay et al., 2018; Peyré & Cuturi, 2019). In Section 4, we report empirical findings based on these divergences as well as on the Wasserstein distance.

## 3 Theoretical properties

In this section, we analyze the properties of the Gaussian-smoothed sliced divergence, in terms of topological and statistical properties and the influence of the Gaussian smoothing parameter $\sigma$ on the distance.

### 3.1 Topology

It has already been shown in Rakotomamonjy & Ralaivola (2021) that the Gaussian-smoothed sliced Wasserstein is a metric on $\mathcal{P}(\mathbb{R}^d)$. In the next, we extend these results to any divergence $\mathrm{D}(\cdot, \cdot)$ under certain assumptions.

**Theorem 3.1.** *For any $\sigma > 0, p \geq 1$, the following properties hold:*

1. *if $\mathrm{D}(\cdot, \cdot)$ is non-negative (or symmetric), then $\mathrm{G}_\sigma \mathrm{SD}_p(\cdot, \cdot)$ is non-negative (or symmetric);*

2. *if $\mathrm{D}(\cdot, \cdot)$ satisfies the identity of indiscernibles, i.e. for $\mu', \nu' \in \mathcal{P}(\mathbb{R})$, $\mathrm{D}(\mu', \nu') = 0$ if and only if $\mu' = \nu'$, then this identity also holds for $\mathrm{G}_\sigma \mathrm{SD}_p(\cdot, \cdot)$ for any $\mu, \nu \in \mathcal{P}_p(\mathbb{R}^d)$;*

3. *if $\mathrm{D}(\cdot, \cdot)$ satisfies the triangle inequality then $\mathrm{G}_\sigma \mathrm{SD}_p(\cdot, \cdot)$ satisfies the triangle inequality.*

The above theorem shows that under mild hypotheses over the base divergence D, as being a metric for instance, the metric property of its Gaussian-smoothed sliced version naturally derives. As exposed in the appendix, the more involved property to prove is the identity of indiscernibles.

We further postponed to the appendix the proofs of the two other topological properties: (i) $\mathrm{G}_\sigma \mathrm{SD}$ metrizes the weak topology on $\mathcal{P}_p(\mathbb{R}^d)$ and (ii) $\mathrm{G}_\sigma \mathrm{SD}$ is lower semi-continuous with respect to the weak topology in $\mathcal{P}_p(\mathbb{R}^d)$.

Now, we establish under which conditions on the divergence D, the convergence of a sequence in $\mathrm{G}_\sigma \mathrm{SD}$ implies weak convergence in $\mathcal{P}_p(\mathbb{R}^d)$. We say that $\{\mu_k\}_{k \in \mathbb{N}}$ *converges weakly* to $\mu$ and write, $\mu_k \Rightarrow \mu$, if $\int f(x) d\mu_k(x) \to \int f(x) d\mu(x)$, as $k \to \infty$, for every $f$ in the space of all bounded continuous real functions.

**Theorem 3.2.** *Let $\sigma > 0, p \geq 1$, $\mu \in \mathcal{P}_p(\mathbb{R}^d)$, and $\{\mu_k \in \mathcal{P}_p(\mathbb{R}^d)\}_{k \in \mathbb{N}}$ a sequence of distributions. Assume that the divergence $\mathrm{D}$ is bounded and metrizes the weak topology on $\mathcal{P}(\mathbb{R})$. Then, $\lim_{k \to \infty} \mathrm{G}_\sigma \mathrm{SD}_p(\mu_k, \mu) = 0$ if and only if $\mu_k \Rightarrow \mu$.*

Note that Theorem 3.2 extends the results of Nadjahi et al. (2020) to Gaussian-smoothed distributions, as we retrieve them as a special case for $\sigma = 0$. In addition, based on Theorem 3.2 by Lin et al. (2021) and the above, we can also claim that the Gaussian-smoothed SWD metrizes the weak convergence.

**Proposition 3.3.** *Let $\sigma > 0, p \geq 1$ and assume that the base divergence D is lower semi-continuous w.r.t. the weak topology in $\mathcal{P}(\mathbb{R})$. Then, $G_\sigma SD_p$ is lower semi-continuous with respect to the weak topology in $\mathcal{P}_p(\mathbb{R}^d)$.*

When the base divergence D is equal to the Wasserstein distance $W_p$, that is lower semi-continuous (Villani, 2009), then Proposition 3.3 shows that the smoothed sliced Wasserstein distance is semi-lower continuous too.

## 3.2 Statistical properties

The next theoretical question we are interested in is about the incurred error when the true distribution $\mu$ is approximated by its empirical distribution $\hat{\mu}_n$. Such a case is common in practical applications where only (high-dimensional) empirical samples are at disposal. Specifically, we are interested in quantifying two key properties of empirical Gaussian-smoothed divergence: *(i)* the convergence of the double empirical $\hat{G}_\sigma SD_p(\hat{\mu}_n, \hat{\nu}_n)$ (see Definition 3.6) to $G_\sigma SD_p(\mu, \nu)$ *(ii)* the convergence of $\widehat{G_\sigma SD}_p(\mu, \nu)$ (see (1)) to $G_\sigma SD_p(\mu, \nu)$, when approximating the expectation over the random projection with sample mean.

Let $\hat{\mu}_n = \frac{1}{n} \sum_{i=1}^{n} \delta_{X_i}$ and $\hat{\nu}_n = \frac{1}{n} \sum_{i=1}^{n} \delta_{Y_i}$ be the empirical probability measures of independent observations. The smoothed Gaussian sliced divergence between $\hat{\mu}_n$ and $\hat{\nu}_n$ is given by

$$G_\sigma SD_p(\hat{\mu}_n, \hat{\nu}_n) = \left( \int_{\mathbb{S}^{d-1}} D^p \left( \mathcal{R}_{\mathbf{u}} \hat{\mu}_n * \mathcal{N}_\sigma, \mathcal{R}_{\mathbf{u}} \hat{\nu}_n * \mathcal{N}_\sigma \right) u_d(\mathbf{u}) d\mathbf{u} \right)^{1/p}.$$

*Remark* 3.4. Remark that for a fixed $\mathbf{u} \in \mathbb{S}^{d-1}$, the distributions $\mathcal{R}_{\mathbf{u}} \hat{\mu}_n * \mathcal{N}_\sigma$ and $\mathcal{R}_{\mathbf{u}} \hat{\nu}_n * \mathcal{N}_\sigma$ are *continuous*, in particular they are a mixture of Gaussian distributions centered on the projected samples with variance $\sigma^2$.

**Lemma 3.5.** *Conditionally on the samples $\{X_i\}_{i=1,\ldots,n}$ and $\{Y_i\}_{i=1,\ldots,n}$, one has: $\mathcal{R}_{\mathbf{u}} \hat{\mu}_n * \mathcal{N}_\sigma = \frac{1}{n} \sum_{i=1}^{n} \mathcal{N}(\mathbf{u}^\top X_i, \sigma^2)$ and $\mathcal{R}_{\mathbf{u}} \hat{\nu}_n * \mathcal{N}_\sigma = \frac{1}{n} \sum_{i=1}^{n} \mathcal{N}(\mathbf{u}^\top Y_i, \sigma^2)$.*

Note that we further need to sample with respect to the continuous mixture Gaussian measures in Lemma 3.5 in order to get a *fully* empirical measure version of $G_\sigma SD(\mu, \nu)$. To this end, we next define the *double empirical divergence* of $G_\sigma SD$.

### 3.2.1 Double empirical divergence of $G_\sigma SD$

Let $T_1^x, \ldots, T_n^x$ and $T_1^y, \ldots, T_n^y$ be i.i.d. observations of $\mathcal{R}_{\mathbf{u}} \hat{\mu}_n * \mathcal{N}_\sigma$ and $\mathcal{R}_{\mathbf{u}} \hat{\nu}_n * \mathcal{N}_\sigma$, respectively. Sampling i.i.d. $\{T_i^x\}_{i=1,\ldots,n}$ is given by the following scheme: for $i = 1, \ldots, n$, we first choose the component $\mathcal{N}(\mathbf{u}^\top X_i, \sigma^2)$ from the mixture $\frac{1}{n} \sum_{i=1}^{n} \mathcal{N}(\mathbf{u}^\top X_i, \sigma^2)$ then we generate $T_i^x = \mathbf{u}^\top X_i + Z_i^x$, where $Z_i^x \sim \mathcal{N}_\sigma$. Hence, we set, for a given $\mathbf{u}$

$$\hat{\hat{\mu}}_n = \frac{1}{n} \sum_{i=1}^{n} \delta_{T_i^x} = \frac{1}{n} \sum_{i=1}^{n} \delta_{\mathbf{u}^\top X_i + Z_i^x} \text{ and } \hat{\hat{\nu}}_n = \frac{1}{n} \sum_{i=1}^{n} \delta_{T_i^y} = \frac{1}{n} \sum_{i=1}^{n} \delta_{\mathbf{u}^\top Y_i + Z_i^y}.$$

The measure $\hat{\hat{\mu}}_n \in \mathcal{P}(\mathbb{R})$ defines an empirical version of the continuous $\mathcal{R}_{\mathbf{u}} \hat{\mu}_n * \mathcal{N}_\sigma$ denoted as $\mathcal{R}_{\mathbf{u}} \widehat{\hat{\mu}_n * \mathcal{N}_\sigma}$ (similarly $\hat{\hat{\nu}}_n = \mathcal{R}_{\mathbf{u}} \widehat{\hat{\nu}_n * \mathcal{N}_\sigma}$). Using the aforementioned notation, we define.

**Definition 3.6.** The double empirical smoothed Gaussian sliced divergence reads as

$$\hat{G}_\sigma SD_p(\hat{\mu}_n, \hat{\nu}_n) \triangleq \left( \int_{\mathbb{S}^{d-1}} D^p(\hat{\hat{\mu}}_n, \hat{\hat{\nu}}_n) u_d(\mathbf{u}) d\mathbf{u} \right)^{1/p}.$$

*Remark* 3.7. *(i)* It is worth to comment the double randomnesses showing in the definition of $\hat{G}_\sigma SD_p(\hat{\mu}_n, \hat{\nu}_n)$: the first comes from sampling according to the original probability measure ($\mu$ or $\nu$) whereas the second takes place from sampling according to the mixture $\frac{1}{n} \sum_{i=1}^{n} \mathcal{N}(\mathbf{u}^\top X_i, \sigma^2)$.
*(ii)* The empirical measure of the convolution $\mathcal{R}_{\mathbf{u}} \widehat{\mu * \mathcal{N}_\sigma}$ could be written as $\frac{1}{n} \sum_{i=1}^{n} \delta_{U_i^x + Q_i^x}$ allowing to

sample *in a one shot* $n$ i.i.d. samples $U_i^x + Q_i^x$ such that $U_i^x \sim \mathcal{R}_{\mathbf{u}}\mu$ and $Q_i^x \sim \mathcal{N}_\sigma$. From an empirical view, sampling according to $\mathcal{R}_{\mathbf{u}}\mu * \mathcal{N}_\sigma$ is intractable. For that reason, our theoretical results and numerical experiments are based on $\hat{\mu}_n, \hat{\nu}_n$, and hence with respect to $\hat{G}_\sigma SD_p(\hat{\mu}_n, \hat{\nu}_n)$.

### 3.2.2 Sample complexity of $G_\sigma SW_p$

Herein, our goal is to quantify the error made when approximating $G_\sigma SW_p(\mu, \nu)$ with $\hat{G}_\sigma SW_p(\hat{\mu}_n, \hat{\nu}_n)$. More precisely, we are interested in establishing an order of the convergence rate of $\hat{G}_\sigma SD_p(\hat{\mu}_n, \hat{\nu}_n)$ towards $G_\sigma SD_p(\mu, \nu)$, according to the sample size $n$. This rate stands for the so-called *sample complexity.*

The convergence results in the sequel are given in expectation. Recall that the empirical distributions are derived from a double sampling process, which leads to consider a double expectations, wrt the origin distribution $\mathbf{E}_{\mu^{\otimes n}}$ and wrt the sampling from the Gaussian smoothing $\mathbf{E}_{\mathcal{N}_\sigma^{\otimes n}}$ where $\mu^{\otimes n}$ and $\mathcal{N}_\sigma^{\otimes n}$ are the $n$-fold product extensions of $\mu$ and $\mathcal{N}_\sigma$, respectively. We first consider the conditional expectation given the samples $X_1, \ldots, X_n$, i.e. $\mathbf{E}_{\mathcal{N}_\sigma^{\otimes n}}[\cdot | X_1, \ldots, X_n]$, and then apply $\mathbf{E}_{\mu^{\otimes n}}$. We denote by

$$\mathbf{E}_{\mu^{\otimes n} | \mathcal{N}_\sigma^{\otimes n}}[\cdot] = \mathbf{E}_{\mu^{\otimes n}}\left[\mathbf{E}_{\mathcal{N}_\sigma^{\otimes n}}[\cdot | X_1, \ldots, X_n]\right].$$

Next, we focus on the sample complexity for the special case of Gaussian-smoothed sliced Wasserstein distance.

**Proposition 3.8.** *Fix $\sigma > 0, p \geq 1$ and $\vartheta > \sqrt{2}$. For $X \sim \mu$, assume that $\int_0^\infty e^{\frac{2\xi^2}{\sigma^2 \vartheta^2}} \mathbf{P}\left[\|X\| > \xi\right] d\xi < \infty$. Then,*

$$\mathbf{E}_{\mu^{\otimes n} | \mathcal{N}_\sigma^{\otimes n}}[\hat{G}_\sigma SW_p(\hat{\mu}_n, \mu)] \leq \Xi_{p,\sigma,\vartheta} \frac{1}{n^{1/2p}} + \Upsilon_{p,\sigma,\mu} \frac{(\log n)^{1/p}}{n^{1/p}},$$

*where $\Xi_{p,\sigma,\vartheta} = \frac{2^{\frac{5}{2}-\frac{5}{4p}}}{\pi^{1/2p}} \sigma^{1-\frac{1}{4p}} \vartheta^{1+\frac{1}{p}} \left(\Gamma\left(p + \frac{1}{2}\right)\left(\sqrt{\frac{4\pi\sigma^2\vartheta^2}{\vartheta^2-2}} + 4\int_0^\infty e^{\frac{2\xi^2}{\sigma^2\vartheta^2}} \mathbf{P}[\|X\| > \xi] d\xi\right)\right)^{1/2p}$ and $\Upsilon_{p,\sigma,\mu} = \frac{2^{2-\frac{2}{2p}} C_p}{\pi^{1/2p}} \sigma^2 \left(\Gamma(p + \frac{1}{2}) \sum_{k=0}^\infty \frac{(-p)_k}{(\frac{1}{2})_k} \frac{(-1)^k}{(2\sigma^2)^k k!} M_{2k}(\mu)\right)^{1/p}$ with $C_p$ is a positive constant depending only on $p$.*

It is worth to note that for $p \in \mathbb{N}^*$, e.g. $p = 2$ (standard choice for numerical experiments), the (pseudo) confluent hypergeometric function $\sum_{k=0}^\infty \frac{(-p)_k}{(\frac{1}{2})_k} \frac{(-1)^k}{(2\sigma^2)^k k!} M_{2k}(\mu)$ is only depending on the $2k$-th moments of $\mu$ for $k = 1, \ldots, p$, since $(-p)_{(k)} = 0$ for $k \geq p+1$. Now, let us sketch the proof of Proposition 3.8: we first insert the proxy term of mixture Gaussian distribution $\frac{1}{n} \sum_{i=1}^n \mathcal{N}(\mathbf{u}\top X_i, \sigma^2)$, then by an application of the triangle inequality on the Wasserstein distance we are faced to control two terms (i) $W_p^p(\hat{\mu}_n, \frac{1}{n} \sum_{i=1}^n \mathcal{N}(\mathbf{u}\top X_i, \sigma^2))$ and (ii) $W_p^p(\frac{1}{n} \sum_{i=1}^n \mathcal{N}(\mathbf{u}\top X_i, \sigma^2), \mu)$. For (i) we get a standard order of $O(\frac{\log n}{n})$, which comes from a by-product of Fournier & Guillin (2015). For (ii), through a coupling via the maximal coupling using the total variation distance (Theorem 6.15 in Villani (2009)), we obtain the order $O(n^{-1/2})$. The control technique for (ii) was inspired from Goldfeld et al. (2020) and Nietert et al. (2021).

*Remark* 3.9. The condition $\int_0^\infty e^{\frac{2\xi^2}{\sigma^2\vartheta^2}} \mathbf{P}\left[\|X\| > \xi\right] d\xi < \infty$ needs $\mathbf{P}\left[\|X\| > \xi\right]$ goes to 0 faster than $e^{-\kappa\xi^2}$ for $\kappa < 2/\sigma^2\vartheta^2$. This can be satisfied when $\|X\|$ is a $\omega$-sub-gausssian ($\omega \geq 0$). Namely, $\mathbf{E}[e^{\eta^\top (X - \mathbf{E}[X])}] \leq e^{\frac{\omega\|\eta\|^2}{2}}$ for all $\eta \in \mathbb{R}^d$. If the parameter $\omega$ verifies $\omega < \sigma\vartheta/2$, then the latter condition holds.

*Remark* 3.10. Note that the sample complexity depends on the amount of smoothing through the moment of the Gaussian noise : the larger the amount of smoothing (and thus the privacy), the worse is the constant of the complexity. Hence, a trade-off on privacy and statistical estimation appears here as a reasonable guarantee on the differential privacy usually requires a large Gaussian variance.

**Proposition 3.11.** *Under the same conditions of Proposition 3.8, we have*

$$\mathbf{E}_{\mu^{\otimes n} | \mathcal{N}_\sigma^{\otimes n}} \mathbf{E}_{\nu^{\otimes n} | \mathcal{N}_\sigma^{\otimes n}}[\hat{G}_\sigma SW_p(\hat{\mu}_n, \hat{\nu}_n)] \leq 3^{1-\frac{1}{p}} G_\sigma SW_p(\mu, \nu) + 3\Xi_{p,\sigma,\vartheta} \frac{1}{n^{1/2p}} + 3^{1-\frac{1}{p}}(\Upsilon_{p,\sigma,\mu} + \Upsilon_{p,\sigma,\nu}) \frac{(\log n)^{1/p}}{n^{1/p}}$$

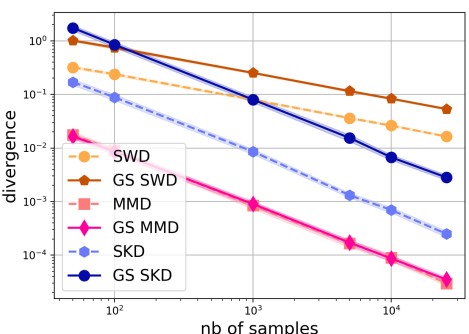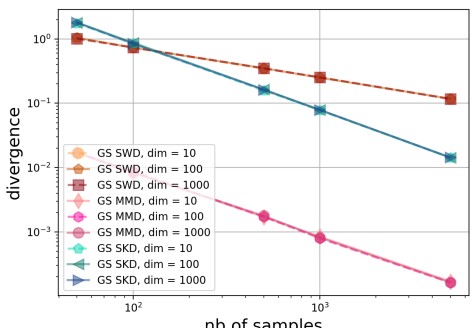

Figure 1: Measuring the divergence between two sets of samples in $\mathbb{R}^{50}$, of increasing size, randomly drawn from $\mathcal{N}(0,\mathbf{I})$. We compare three sliced divergences and their Gaussian-smoothed sliced versions with a $\sigma = 3$: (top) dimension has been set to $d = 50$; (bottom) sample complexity with different dimensions. This plot confirms that the complexity is dimension-independent.

*and*

$$\mathrm{G}_\sigma\mathrm{SW}_p(\mu,\nu) \le 3^{1-\frac{1}{p}}\mathbf{E}_{\mu^{\otimes n}|\mathcal{N}_\sigma^{\otimes n}}\mathbf{E}_{\nu^{\otimes n}|\mathcal{N}_\sigma^{\otimes n}}[\hat{\mathrm{G}}_\sigma\mathrm{SW}_p(\hat{\mu}_n,\hat{\nu}_n)] + 3\Xi_{p,\sigma,\vartheta}\frac{1}{n^{1/2p}} + 3^{1-\frac{1}{p}}(\Upsilon_{p,\sigma,\mu} + \Upsilon_{p,\sigma,\nu})\frac{(\log n)^{1/p}}{n^{1/p}}.$$

Proof of Proposition 3.11 relies on a double application of triangle inequality satisfied by Wasserstein distance as follows: $\mathrm{W}_p(\hat{\hat{\mu}}_n,\hat{\hat{\nu}}_n) \le \mathrm{W}_p(\hat{\hat{\mu}}_n,\mathcal{R}_\mathbf{u}\mu * \mathcal{N}_\sigma) + \mathrm{W}_p(\mathcal{R}_\mathbf{u}\mu * \mathcal{N}_\sigma, \mathcal{R}_\mathbf{u}\nu * \mathcal{N}_\sigma) + \mathrm{W}_p(\mathcal{R}_\mathbf{u}\nu * \mathcal{N}_\sigma, \hat{\hat{\nu}}_n)$, combined with Proposition 3.8. This gives a non sharp convergence result since we get the constant $3^{1-\frac{1}{p}}$ in front of $\mathbf{E}_{\mu^{\otimes n}|\mathcal{N}_\sigma^{\otimes n}}\mathbf{E}_{\nu^{\otimes n}|\mathcal{N}_\sigma^{\otimes n}}[\hat{\mathrm{G}}_\sigma\mathrm{SW}_p(\hat{\mu}_n,\hat{\nu}_n)]$ or $\mathrm{G}_\sigma\mathrm{SW}_p(\mu,\nu)$. However, when the power $p = 1$ we obtain a sharp convergence result with $O(n^{-1/2})$, namely

$$\mathbf{E}_{\mu^{\otimes n}|\mathcal{N}_\sigma^{\otimes n}}\mathbf{E}_{\nu^{\otimes n}|\mathcal{N}_\sigma^{\otimes n}}[|\hat{\mathrm{G}}_\sigma\mathrm{SW}(\hat{\mu}_n,\hat{\nu}_n) - \mathrm{G}_\sigma\mathrm{SW}(\mu,\nu)|] \le 3\Xi_{1,\sigma,\vartheta}\frac{1}{\sqrt{n}} + (\Upsilon_{1,\sigma,\mu} + \Upsilon_{1,\sigma,\nu})\frac{\log n}{n}$$

Despite that our theoretical results hold only for Gaussian-smoothed sliced Wasserstein distance, our empirical results show that given other base divergences D, shows that the sample complexity of $\mathrm{G}_\sigma\mathrm{SD}^p$ is proportional to the one dimensional sample complexity of $\mathrm{D}^p$ $(p = 2)$. Figure 1 provides an empirical illustration of this statement.

### 3.2.3 Projection complexity

To compute the Gaussian-smoothed sliced divergence, one may resort to a Monte Carlo scheme to numerically approximate the integral in $\mathrm{G}_\sigma\mathrm{SD}_p(\mu,\nu)$. Towards this, let define the following sum:

$$\widehat{\mathrm{G}_\sigma\mathrm{SD}}_p(\mu,\nu) = \left(\frac{1}{L}\sum_{l=1}^{L}\mathrm{D}_p(\mathcal{R}_{\mathbf{u}_l}\mu * \mathcal{N}_\sigma, \mathcal{R}_{\mathbf{u}_l}\nu * \mathcal{N}_\sigma)\right)^{1/p}, \tag{1}$$

where $\mathbf{u}_l$ is a random vector uniformly drawn from $\mathbb{S}^{d-1}$, for $l = 1,\ldots,L$. Theorem 3.12 shows that for a fixed dimension $d$, the root mean square error of Monte Carlo (MC) approximation is of order $O\left(\frac{1}{\sqrt{L}}\right)$, which corresponds to the projection complexity. We denote by $u_d^{\otimes L}$ and the $L$-fold product extensions of the uniform measure $u_d$ on the unit sphere.

**Proposition 3.12.** *Let $\sigma > 0, p \ge 1$. Then the error related to the MC-estimation of $\mathrm{G}_\sigma\mathrm{SD}_p$ is bounded as follows*

$$\mathbf{E}_{u_d^{\otimes L}}[|\widehat{\mathrm{G}_\sigma\mathrm{SD}}_p^{\ p}(\mu,\nu) - \mathrm{G}_\sigma\mathrm{SD}_p^p(\mu,\nu)|] \le \frac{A(p,\sigma)}{\sqrt{L}},$$

*where* $A^2(p,\sigma) = \int_{\mathbb{S}^{d-1}} \big( \mathrm{D}^p(\mathcal{R}_{\mathbf{u}}\mu * \mathcal{N}_\sigma, \mathcal{R}_{\mathbf{u}}\nu * \mathcal{N}_\sigma) - \bar{\tau}_p \big)^2 u_d(\mathbf{u})\mathrm{d}\mathbf{u}$, *with* $\bar{\tau}_p = \int_{\mathbb{S}^{d-1}} \mathrm{D}^p(\mathcal{R}_{\mathbf{u}}\mu * \mathcal{N}_\sigma, \mathcal{R}_{\mathbf{u}}\nu * \mathcal{N}_\sigma)u_d(\mathbf{u})\,d\mathbf{u}$.

The term $A^2(p,\sigma)$ corresponds to the variance of $\mathrm{D}^p(\mathcal{R}_{\mathbf{u}}\mu * \mathcal{N}_\sigma, \mathcal{R}_{\mathbf{u}}\nu * \mathcal{N}_\sigma)$ with respect to $\mathbf{u} \sim u_d$. It is worth to note that the precision of the Monte Carlo scheme approximation depends on the number of projections $L$ and the variance of the evaluations of the divergence $\mathrm{D}^p$. The estimation error decreases at the rate $L^{-1/2}$ according to the number of projections used to compute the smoothed sliced divergence.

Given the above results, we provide a finer analysis of $\mathrm{G}_\sigma\mathrm{SW}_p(\mu,\nu)$'s sample complexity. Towards this ends, for a fixed random projection $\mathbf{u}_l, (1 \le l \le L)$ we define $\hat{\mu}_{n,l} = \frac{1}{n}\sum_{i=1}^n \delta_{\mathbf{u}_l^\top X_i + Z_i^x}$ (similarly for $\hat{\nu}_{n,l}$) and set

$$\widehat{\mathrm{G}_\sigma\mathrm{SD}}_p(\hat{\mu}_n,\hat{\nu}_n) = \Big( \frac{1}{L}\sum_{l=1}^L \mathrm{W}_p^p(\hat{\mu}_{n,l}, \hat{\nu}_{n,l}) \Big)^{1/p}$$

The overall complexity of $\mathrm{G}_\sigma\mathrm{SD}_p(\mu,\nu)$ consists in its approximation by sampling and projection of the origin probability measures $\mu,\nu$, i.e. through $\widehat{\mathrm{G}_\sigma\mathrm{SD}}_p(\hat{\mu}_n,\hat{\nu}_n)$. By application of triangle inequality, one has

$$|\widehat{\mathrm{G}_\sigma\mathrm{SW}}_p^{\,p}(\hat{\mu}_n,\hat{\nu}_n) - \mathrm{G}_\sigma\mathrm{SW}_p^p(\mu,\nu)| \le |\widehat{\mathrm{G}_\sigma\mathrm{SW}}_p^{\,p}(\hat{\mu}_n,\hat{\nu}_n) - \hat{\mathrm{G}}_\sigma\mathrm{SW}_p^p(\hat{\mu}_n,\hat{\nu}_n)| + |\hat{\mathrm{G}}_\sigma\mathrm{SW}_p^p(\hat{\mu}_n,\hat{\nu}_n) - \mathrm{G}_\sigma\mathrm{SW}_p(\mu,\nu)|.$$

The first term in the right-hand-side (RHS) of the latter decomposition can be controlled by Proposition 3.12 in the following way:

$$\mathbf{E}_{u_d^{\otimes L}}\big[|\widehat{\mathrm{G}_\sigma\mathrm{SW}}_p^{\,p}(\hat{\mu}_n,\hat{\nu}_n) - \hat{\mathrm{G}}_\sigma\mathrm{SW}_p^p(\hat{\mu}_n,\hat{\nu}_n)|\big] \le \frac{\hat{A}(p,\sigma)}{\sqrt{L}} \triangleq \frac{\{\mathbf{V}_{\mathbf{u}\sim u_d}[\mathrm{W}_p^p(\hat{\mu}_n,\hat{\nu}_n)]\}^{1/2}}{\sqrt{L}}.$$

However we don't have a proper control for $p \ge 2$ of the second term in the RHS, $|\hat{\mathrm{G}}_\sigma\mathrm{SW}_p^p(\hat{\mu}_n,\hat{\nu}_n) - \mathrm{G}_\sigma\mathrm{SW}_p(\mu,\nu)|$, as it can be seen from Proposition 3.11. For that reason, we derive an overall complexity in the case of $p = 1$.

**Corollary 3.13.** *The sample and projection complexities of* $\mathrm{G}_\sigma\mathrm{SW}(\mu,\nu)$ *reads as* $\mathrm{complexity}(\mathrm{G}_\sigma\mathrm{SW}) = O(n^{-1/2} + L^{-1/2})$. *If we consider the number of projections as* $L = \lfloor n^\beta \rfloor$ *for some* $\beta \in (0,1)$ *then the overall complexity* $\mathrm{complexity}(\mathrm{G}_\sigma\mathrm{SW}(\mu,\nu)) = O(n^{-\beta/2})$.

### 3.3 Noise-level dependencies

The parameter $\sigma$ of the Gaussian smoothing function $\mathcal{N}_\sigma$ may significantly influence the attained privacy level. Hence, we provide theoretical results analyzing the effect of the noise level $\sigma$ on the induced Gaussian-smoothed sliced divergence.

### 3.4 Order relation

We first show that the noise level tends to reduce the difference between two distributions as measured using $\mathrm{G}_\sigma\mathrm{SD}^p(\mu,\nu)$ provided the base divergence $D$ satisfies some mild assumptions.

**Proposition 3.14.** *Let* $\mu,\nu \in \mathcal{P}_p(\mathbb{R}^d)$ *and consider the noise levels* $\sigma_1,\sigma_2$ *such that* $0 \le \sigma_1 \le \sigma_2 < \infty$. *Assume that the base divergence* $\mathrm{D}$ *satisfies* $\mathrm{D}(\mu' * \mathcal{N}_{\sigma_2}, \nu' * \mathcal{N}_{\sigma_2}) \le \mathrm{D}(\mu' * \mathcal{N}_{\sigma_1}, \nu' * \mathcal{N}_{\sigma_1})$, *for any* $\mu',\nu' \in \mathcal{P}(\mathbb{R})$. *Then,* $\mathrm{G}_{\sigma_2}\mathrm{SD}^p(\mu,\nu) \le \mathrm{G}_{\sigma_1}\mathrm{SD}^p(\mu,\nu)$.

Note that the assumption for the base divergence inequality holds for the Gaussian-smoothed Wasserstein distance Nietert et al. (2021). While we conjecture that it holds also for smoothed Sinkhorn and MMD, we leave the proofs for future works. Based on the property in Proposition 3.14, we show some specific properties of the metric with respect to the noise level $\sigma$.

**Proposition 3.15.** $\mathrm{G}_\sigma\mathrm{SD}^p(\mu,\nu)$ *is decreasing with respect to* $\sigma$ *and we have* $\lim_{\sigma\to 0}\mathrm{G}_\sigma\mathrm{SD}^p(\mu,\nu) = \mathrm{D}^p(\mu,\nu)$.

The proof of Proposition 3.15 comes straightforwardly from Proposition 3.14 by taking $\sigma_2 = \sigma$ and letting $\sigma_1 \to 0$. This property interestingly states that the $\mathrm{G}_\sigma\mathrm{SD}^p$ recovers the sliced divergence when the noise level vanishes. We end up this section by providing a relation between Gaussian-smoothed sliced Wasserstein distances under two noise levels.

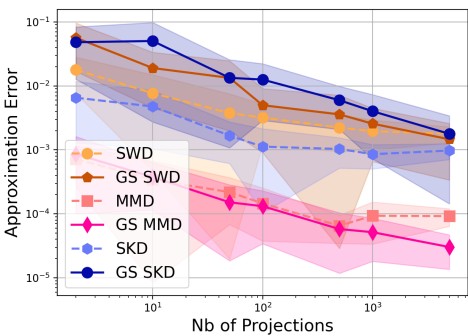

Figure 2: Absolute difference between the approximated Monte Carlo approximation of all divergences compared to the true one (evaluated with $10,000$ number of projections). The two sets of $500$ samples in $\mathbb{R}^{50}$ are randomly drawn from $\mathcal{N}(0, \mathbf{I})$. The Gaussian-smoothed sliced divergences are parameterized with $\sigma = 3$.

**Proposition 3.16.** *Let $0 \leq \sigma_1 \leq \sigma_2$ be two noise levels. Then, one has $\mathrm{G}_{\sigma_2}\mathrm{SW}_p(\mu, \nu) \leq \mathrm{G}_{\sigma_1}\mathrm{SW}_p(\mu, \nu)$ and*

$$|\mathrm{G}_{\sigma_1}\mathrm{SW}_p(\mu, \nu) - \mathrm{G}_{\sigma_2}\mathrm{SW}_p(\mu, \nu)| \leq (2^{1-\frac{1}{p}} - 1)\,\mathrm{G}_{\sigma_2}\mathrm{SW}_p(\mu, \nu) + 2^{\frac{5}{2}}(\sigma_2^2 - \sigma_1^2),$$

*in particular for $p = 1$, $|\mathrm{G}_{\sigma}\mathrm{SW}(\mu, \nu) - \mathrm{G}_{\sigma_2}\mathrm{SW}(\mu, \nu)| \leq 2^{\frac{5}{2}}(\sigma_2^2 - \sigma_1^2)$.*

### 3.4.1 Continuity

Now we analyze the continuity properties of some $\mathrm{G}_{\sigma}\mathrm{SD}^p(\mu, \nu)$ w.r.t. the noise level.

**Proposition 3.17.** *For any two distributions $\mu$ and $\nu$ for which the sliced Wasserstein is well-defined, the Gaussian-smoothed sliced Wasserstein distance is continuous w.r.t. to $\sigma$.*

**Proposition 3.18.** *Assume that the kernel defining the maximum mean discrepancy* (MMD) *divergence is bounded. Then the Gaussian-smoothed sliced $\mathrm{G}_{\sigma}\mathrm{MMD}$ is continuous w.r.t. to $\sigma$.*

The above propositions show that most distribution divergences are continuous with respect to $\sigma$ under mild conditions.

## 4 Numerical Experiments

In this section, we report on a series of experiments that support the established theoretical results. We also highlight the usefulness of the findings in a context of privacy-preserving domain adaptation problem.

### 4.1 Supporting the theoretical results

**Sample complexity.** The first experiment (see Figure 1) analyzes the sample complexity of different base divergences. It shows that the sample complexity stays similar to the one of their original and sliced counterparts up to a constant (see Proposition 3.8). For this purpose, we have considered samples in $\mathbb{R}^d$ randomly drawn from a Normal distribution $\mathcal{N}(0, \mathbf{I})$. For the Sinkhorn divergence, the entropy regularization has been set to 0.1 and for MMD, we used a Gaussian kernel for which the bandwidth has been set to the mean of all pairwise distances between samples. The number of projections has been fixed to $L = 50$ and we perform 20 runs per experiment. For the first study, the convergence rate has been evaluated by increasing the samples number up to 25,000 with fixed dimension $d = 50$. For the second one, we vary both the dimension and the number of samples.

Figure 1 shows the sample complexity of some sliced divergences, respectively noted as SWD, SKD and MMD for Sliced Wasserstein distance, Sinkhorn divergence and Maximum Mean discrepancy and their Gaussian-smoothed sliced versions, named as GS SWD, GS SKD and GS MMD. On the top plot, we can see that all Gaussian-smoothed sliced divergences preserve the complexity rate with just a slight to moderate

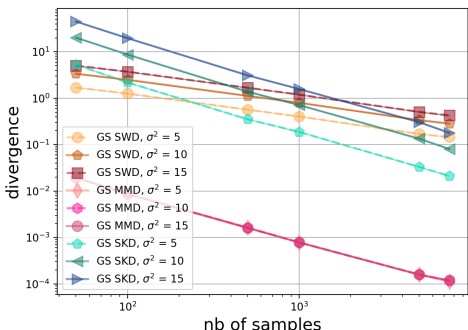

Figure 3: Measuring the divergence between two sets of samples in $\mathbb{R}^{50}$ drawn from $\mathcal{N}(0, \mathbf{I})$. We plot the sample complexity for different Gaussian-smoothed sliced divergence at different level of noises.

overhead. The worst difference is for Sinkhorn divergence, while MMD almost comes for free in term of complexity. From the bottom plot where sample complexities for different dimensions $d$ are given, we confirm the finding that Gaussian smoothing keeps the independence of the convergence rate to the dimension of sliced divergences.

Two other experiments on the sample complexity and identity of indiscernibles are also reported in the supplementary material.

**Projection complexity.** We have also investigated the impact of the number of projections when estimating the distance between two sets of 500 samples drawn from the same distribution, $\mathcal{N}(0, \mathbf{I})$. Figure 2 plots the approximation error between the true expectation of the sliced divergences (computed for a number of $L = 10,000$ projections) and its approximated versions. We remark that, for all methods, the error ranges within 10-fold when approximating with 50 projections and decreases with the number of projections.

**Performance path on the impact of the noise parameter.** Since the Gaussian smoothing parameter $\sigma$ is key in a privacy preserving context, as it impacts on the level of privacy of the Gaussian mechanism, we have analyzed its impact on the smoothed sliced divergence. We have reproduced the experiment for the sample complexity but with different values of $\sigma$. The number of projections has been set to 50. Figure 3 shows these sample complexities. The first very interesting point to note is that the smoothing parameter has almost no effect on the GS MMD sample complexity. For the GS SWD and GS SKD divergences, instead, the smoothing tends to increase the divergence at fixed number of samples. Another interpretation is that to achieve a given value of divergence, one needs more far samples when the smoothing is larger (*i.e.* for getting a given divergence value at $\sigma = 5$, one needs almost 10-fold more samples for $\sigma = 15$). This overhead of samples needed when smoothing increases is properly described, for the Gaussian-smoothed sliced SWD in our Proposition 3.8, as the sample complexity depends on the moments of the Gaussian.

As for conclusion from these analyses, we highlight that the Gaussian-smoothed sliced MMD seems to present several strong benefits: its sample complexity does not depend on the dimension and seems to be the best one among the divergence we considered. More interestingly, it is not impacted by the amount of Gaussian smoothing and thus not impacted by a desired privacy level.

## 4.2 Domain adaptation with $\mathrm{G}_\sigma \mathrm{SW}$

As an application, we have considered the problem of unsupervised domain adaptation for a classification task. In this context, given source examples $\mathbf{X}_s$ and their label $\mathbf{y}_s$ and unlabeled target examples $\mathbf{X}_t$, our goal is to design a classifier $h(\cdot)$ learned from the source examples that generalizes well on the target ones. A classical approach consists in learning a representation mapping $g(\cdot)$ that leads to invariant latent representations, invariance being measured as a distance between empirical distributions of mapped source and target samples.

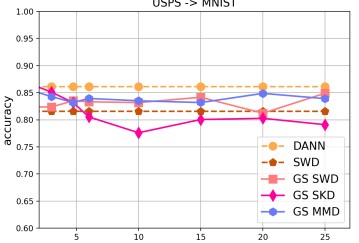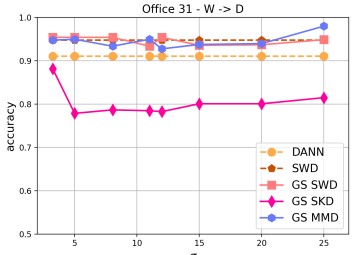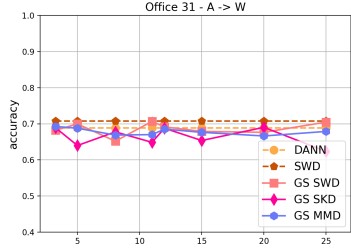

Figure 4: Domain adaptation performances using different divergences on distributions with respect to the Gaussian smoothing. (Left) USPS to MNIST. (Middle) Office-31 Webcam to DSLR. (Right) Office-31 Amazon to Webcam.

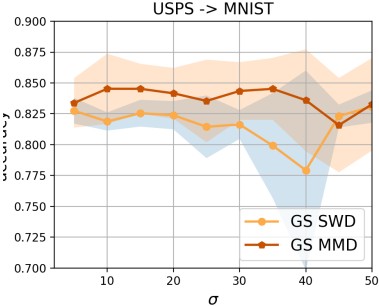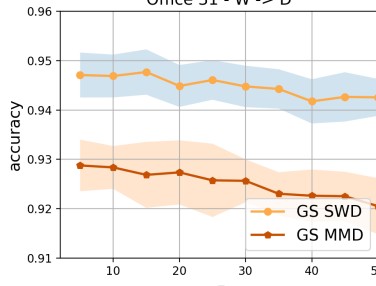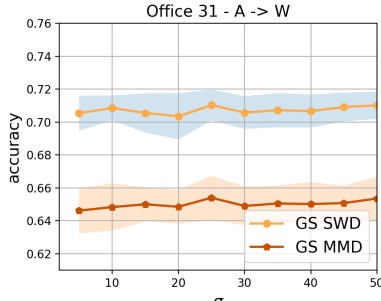

Figure 5: Domain adaptation performances using different divergences on distributions with respect to the Gaussian smoothing using **one-epoch-fine-tuned** models. (Left) USPS to MNIST. (Middle) Office-31 Webcam to DSLR. (Right) Office-31 Amazon to Webcam.

Formally, this leads to the following problem

$$\min_{g,h} \left\{ \mathcal{L}_c(h(g(\mathbf{X}_s)), \mathbf{y}_s) + \mathcal{D}(g(\mathbf{X}_s), g(\mathbf{X}_t)) \right\}$$

where $\mathcal{L}_c$ can be the cross-entropy loss or a quadratic loss and $\mathcal{D}$ a divergence between empirical distributions, in our case, $\mathcal{D}$ will be any Gaussian-smoothed sliced divergence. We solve this problem through stochastic gradient descent, similarly to many approaches that use sliced Wasserstein distance as a distribution distance Lee et al. (2019). Note that, in practice, using a smoothed divergence preserves the privacy of the target samples as shown by (Rakotomamonjy & Ralaivola, 2021).

When performing such model adaptation, a privacy/utility trade-off that has to be handled. In practice, one would prefer the most private model while not hurting its performance. Hence, one would seek the largest noise level $\sigma > 0$ to use while preserving accuracy on target domain. Hence, it is useful to evaluate how the model performs on a range of noise level (hence, privacy level). This can be computationally expensive at it requires to fully train several models on hundreds of epochs. Instead, we leverage on the continuity of our $\mathrm{G}_\sigma\mathrm{SD}$ to employ a fine-tuning strategy: we train a domain adaptation model for the largest desired value of $\sigma$ (over the full number of epochs) and when $\sigma$ is decreased, we just fine-tune the lasted model by training on only one epoch.

Our experiments evaluate the studied Gaussian-smoothed sliced divergences in classical unsupervised domain adaptation. We have considered two datasets: a handwritten digit recognition (USPS/MNIST) and Office 31 datasets.

In our first analysis, we have compared our $\mathrm{G}_\sigma\mathrm{SD}$ performances with non-smoothed divergences. The first one is the sliced Wasserstein distance (SWD) Lee et al. (2019) and the second one is the Jenssen-Shannon approximation based on adversarial approach, known as DANN Ganin & Lempitsky (2015). For all methods

and for each dataset, we used the same neural network architecture for representation mapping and for classification. Approaches differ only on how distance between distributions have been computed. Here for each noise value $\sigma$, we have trained the model from scratch for 100 epochs. Results are depicted in Figure 4. For the two problems, we can see that performances obtained with the Gaussian-smoothed sliced Wasserstein or MMD divergences are similar to those obtained with DANN or SWD across all ranges of noise. The smoothed version of Sinkhorn is less stable and induces a slight loss of performance. Owing to the metric property and the induced weak topology, the privacy preservation comes almost without loss of performance in this domain adaptation context.

In the second analysis, we have studied the privacy/utility trade-off when fine-tuning models, using only one epoch, for decreasing values of $\sigma$. Results are shown in Figure 5. They highlight that depending on the data and the used smoothed divergence, performance varies between one percent for Office 31 to four percent for USPS to MNIST. Note that except for the largest value of $\sigma$, we are training a model using only one epoch instead of a hundred. A very large gain in complexity is thus achieved for swiping the full range of noise level. Hence depending on the importance this slight drop in performance will have, it is worth using a large value of $\sigma$ and preserving strong privacy or go through a validation procedure of several (cheaply obtained) models.

## 5 Conclusion

This work provided the properties of Gaussian-smoothed sliced divergences for comparing distributions. We derived several theoretical results related to their topological and statistical properties and showed, under mild conditions on their base divergences, the smoothing and slicing operations preserves the metric property. From a statistical point of view, we introduced the double empirical distribution and focused on the sample complexity of the smoothed sliced Wasserstein distance and we proved that it converges with a rate $O(n^{-1/2p})$. We furhter analyzed the behavior of these divergences on domain adaptation problems and confirm the fact that using those divergences yields only to slight loss of performances while preserving privacy. Note that in the obtained bound we use upper bound of higher moments of the smoothing distribution. An important direction for future research is considering non Gaussian smoothing distribution enjoying this property.

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

## A  Proofs

In the following sections, we give the proofs of the theoretical guarantees given in the main of the paper.

### A.1  Proof of Theorem 3.1: $\mathrm{G}_\sigma\mathrm{SD}_p$ is a proper metric on $\mathcal{P}_p(\mathbb{R}^d) \times \mathcal{P}_p(\mathbb{R}^d)$

Before starting the proof, we add this notation: the characteristic function of a probability distribution $\mu \in \mathcal{P}(\mathbb{R}^d)$ is $\varphi_\mu(t) = \mathbf{E}_\mu[e^{iX^\top t}]$. Given this definition, similarly to the Fourier transform, the characteristic function of the convolution of two probability distributions readsas $\varphi_{\nu*\mu}(t) = \varphi_\nu(t) \cdot \varphi_\mu(t)$.

• *Non-negativity (or symmetry).* The non-negativity (or symmetry) follows directly from the non-negativity (or symmetry) of $\mathrm{D}^p$, see Definition 2.3.

• *Identity property.* If the base divergence $\mathrm{D}^p$ satisfies the identity property in one dimensional measures, then for any $\mu \in \mathcal{P}_p(\mathbb{R}^d)$ and $\mathbf{u} \in \mathbb{S}^{d-1}$, one has that $\mathrm{D}_p(\mathcal{R}_{\mathbf{u}}\mu * \mathcal{N}_\sigma, \mathcal{R}_{\mathbf{u}}\mu * \mathcal{N}_\sigma) = 0$, hence, by Definition 2.3, $\mathrm{G}_\sigma\mathrm{SD}_p(\mu,\mu) = 0$. Let us now prove the fact that for any $\mu,\nu \in \mathcal{P}_p(\mathbb{R}^d), \mathrm{G}_\sigma\mathrm{SD}^p(\mu,\nu) = 0$ entails $\mu = \nu$ a.s. On one hand, $\mathrm{G}_\sigma\mathrm{SD}_p(\mu,\nu) = 0$ gives the fact that $\mathrm{D}_p(\mathcal{R}_{\mathbf{u}}\mu * \mathcal{N}_\sigma, \mathcal{R}_{\mathbf{u}}\nu * \mathcal{N}_\sigma) = 0$ for $u_d$-almost every $\mathbf{u} \in \mathbb{S}^{d-1}$, hence $\mathcal{R}_{\mathbf{u}}\mu * \mathcal{N}_\sigma = \mathcal{R}_{\mathbf{u}}\nu * \mathcal{N}_\sigma$ for $u_d$-almost every $\mathbf{u} \in \mathbb{S}^{d-1}$. Following the techniques in proof of Proposition 5.1.2 in Bonnotte (2013), for any measure $\eta \in \mathcal{P}(\mathbb{R}^m)$ (with $m \geq 1$), $\mathcal{F}[\eta](\cdot)$ stands for the Fourier transform of $\eta$ and is given as $\mathcal{F}[\eta](\mathbf{v}) = \int_{\mathbb{R}^m} e^{-i\mathbf{s}^\top \mathbf{v}} \mathrm{d}\eta(\mathbf{s})$ for any $\mathbf{v} \in \mathbb{R}^m$. Then

$$
\begin{aligned}
\mathcal{F}[\mathcal{R}_{\mathbf{u}}\mu * \mathcal{N}_\sigma](v) &= \int_{\mathbb{R}} e^{-ivt} \mathrm{d}(\mathcal{R}_{\mathbf{u}}\mu * \mathcal{N}_\sigma)(t) \\
&= \int_{\mathbb{R}} \int_{\mathbb{R}} e^{-i(r+t)v} \mathrm{d}\mathcal{R}_{\mathbf{u}}\mu(r) \mathrm{d}\mathcal{N}_\sigma(t) \quad \text{(by the definition of the convolution operator)} \\
&= \int_{\mathbb{R}^d} \int_{\mathbb{R}} e^{-i(\langle\mathbf{u},\mathbf{s}\rangle+t)v} \mathrm{d}\mu(\mathbf{s}) \mathrm{d}\mathcal{N}_\sigma(t) \quad \text{(by the definition of Radon Transform)} \\
&= \int_{\mathbb{R}} e^{-itv} \mathrm{d}\mathcal{N}_\sigma(t) \int_{\mathbb{R}^d} e^{-i(\langle\mathbf{u},s\rangle)v} \mathrm{d}\mu(\mathbf{s}) \\
&= \mathcal{F}[\mathcal{N}_\sigma](v)\mathcal{F}[\mu](v\mathbf{u}).
\end{aligned}
$$

Since for $u_d$-almost every $\mathbf{u} \in \mathbb{S}^{d-1}, \mathcal{R}_{\mathbf{u}}\mu * \mathcal{N}_\sigma = \mathcal{R}_{\mathbf{u}}\nu * \mathcal{N}_\sigma$, and hence $\mathcal{F}[\mathcal{R}_{\mathbf{u}}\mu * \mathcal{N}_\sigma] = \mathcal{F}[\mathcal{R}_{\mathbf{u}}\nu * \mathcal{N}_\sigma] \Leftrightarrow \mathcal{F}[\mathcal{N}_\sigma]\mathcal{F}[\mu] = \mathcal{F}[\mathcal{N}_\sigma]\mathcal{F}[\nu]$ (by the Fourier transform of the convolution) $\Leftrightarrow \mathcal{F}[\mu] = \mathcal{F}[\nu]$. Since the Fourier transform is injective, we conclude that $\mu = \nu$.

• *Triangle inequality.* Assume that D is a metric and let $\mu,\nu,\eta \in \mathcal{P}_p(\mathbb{R}^d)$. We then have

$$
\begin{aligned}
\mathrm{G}_\sigma\mathrm{SD}_p(\mu,\nu) &= \left( \int_{\mathbb{S}^{d-1}} \mathrm{D}^p(\mathcal{R}_{\mathbf{u}}\mu * \mathcal{N}_\sigma, \mathcal{R}_{\mathbf{u}}\nu * \mathcal{N}_\sigma) u_d(\mathbf{u})\mathrm{d}\mathbf{u} \right)^{1/p} \\
&\leq \left( \int_{\mathbb{S}^{d-1}} \left( \mathrm{D}(\mathcal{R}_{\mathbf{u}}\mu * \mathcal{N}_\sigma, \mathcal{R}_{\mathbf{u}}\eta * \mathcal{N}_\sigma) + \mathrm{D}(\mathcal{R}_{\mathbf{u}}\eta * \mathcal{N}_\sigma, \mathcal{R}_{\mathbf{u}}\nu * \mathcal{N}_\sigma) \right)^p u_d(\mathbf{u})\mathrm{d}\mathbf{u} \right)^{1/p} \\
&\underset{(\star)}{\leq} \left( \int_{\mathbb{S}^{d-1}} \left( \mathrm{D}^p(\mathcal{R}_{\mathbf{u}}\mu * \mathcal{N}_\sigma, \mathcal{R}_{\mathbf{u}}\eta * \mathcal{N}_\sigma) u_d(\mathbf{u})\mathrm{d}\mathbf{u} \right)^{1/p} + \left( \int_{\mathbb{S}^{d-1}} \mathrm{D}^p(\mathcal{R}_{\mathbf{u}}\eta * \mathcal{N}_\sigma, \mathcal{R}_{\mathbf{u}}\nu * \mathcal{N}_\sigma) \right)^p u_d(\mathbf{u})\mathrm{d}\mathbf{u} \right)^{1/p} \\
&= \mathrm{G}_\sigma\mathrm{SD}_p(\mu,\eta) + \mathrm{G}_\sigma\mathrm{SD}_p(\eta,\nu),
\end{aligned}
$$

where inequality in $(\star)$ follows from the application of Minkowski inequality.

### A.2  Proof of Theorem 3.2: $\mathrm{G}_\sigma\mathrm{SD}_p$ metrizes the weak topology

The proof is done by double implications and the technical material relies on the continuous mapping theorem (Athreya & Lahiri, 2006) and bounded convergence theorem for the first direct implication "$\Rightarrow$". The second one, "$\Leftarrow$", is based on the fact that weak convergence is equivalent to the convergence corresponding to Lévy-Prokhorov distance (Huber, 2011)

"$\Rightarrow$" Assume that $\mu_k \Rightarrow \mu$. Fix $\mathbf{u} \in \mathbb{S}^{d-1}$, the mapping $\mathbf{u} \mapsto \mathcal{R}_{\mathbf{u}}$ is continuous from $\mathbb{R}^d$ to $\mathbb{R}$, then an application of continuous mapping theorem (Athreya & Lahiri, 2006) entails that $\mathcal{R}_{\mathbf{u}}\mu_k \Rightarrow \mathcal{R}_{\mathbf{u}}\mu$. By Lévy's continuity

theorem (Athreya & Lahiri, 2006) $\mathcal{R}_{\mathbf{u}}\mu_k * \mathcal{N}_\sigma \Rightarrow \mathcal{R}_{\mathbf{u}}\mu * \mathcal{N}_\sigma$. Therefore, $\lim_{k\to\infty} \mathrm{D}(\mathcal{R}_{\mathbf{u}}\mu_k, \mathcal{R}_{\mathbf{u}}\mu * \mathcal{N}_\sigma) = 0$. Since we suppose that the divergence D is bounded, then there exists $K \geq 0$ such that for any $k$, $\mathrm{D}^p(\mathcal{R}_{\mathbf{u}}\mu_k, \mathcal{R}_{\mathbf{u}}\mu * \mathcal{N}_\sigma) \leq K$. An application of bounded convergence theorem yields

$$\lim_{k\to\infty} \mathrm{G}_\sigma \mathrm{SD}_p(\mu_k, \mu) = \Big( \int_{\mathbb{S}^{d-1}} \lim_{k\to\infty} \mathrm{D}^p(\mathcal{R}_{\mathbf{u}}\mu_k * \mathcal{N}_\sigma, \mathcal{R}_{\mathbf{u}}\mu * \mathcal{N}_\sigma) u_d(\mathbf{u})\mathrm{d}\mathbf{u} \Big)^{1/p} = 0.$$

"$\Leftarrow$" (By contrapositive). Suppose that $\mu_k$ doesn't converge weakly to $\mu$ and assume that $\lim_{k\to\infty} \mathrm{G}_\sigma \mathrm{SD}^p(\mu_k, \mu) = 0$. On one hand, since $\mathbb{R}^d$ is a complete separable space then the weak convergence is equivalent to the convergence corresponding to Lévy-Prokhorov distance $\Lambda$ defined as: The Lévy-Prokhorov distance $\Lambda(\eta, \zeta)$ between $\eta, \zeta \in \mathscr{P}((E, \rho), \mathcal{T})$ (space of probability measures on a measurable metric space) is given by:

$$\Lambda(\eta, \zeta) = \inf_{\varepsilon > 0}\{\eta(A) < \zeta(A^\varepsilon) + \varepsilon, \quad \zeta(A) < \eta(A^\varepsilon) + \varepsilon, \quad \text{for all } A \in \mathcal{T}\}, \text{ where } A^\varepsilon = \{x \in E : \rho(x, A) < \varepsilon\}.$$

Hence there exists $\varepsilon > 0$ and a subsequence $\{\mu_{s(k)}\}_{k\in\mathbb{N}}$ such that $\Lambda(\mu_{s(k)}, \mu) > \varepsilon$. One the other hand, we have $\lim_{k\to\infty} \mathrm{G}_\sigma \mathrm{SD}^p(\mu_{s(k)}, \mu) = 0$, that is equivalent to $\{\mathrm{D}(\mathcal{R}_{\mathbf{u}}\mu_{s(k)} * \mathcal{N}_\sigma, \mathcal{R}_{\mathbf{u}}\nu * \mathcal{N}_\sigma)\}_k$ converges to 0 in $L^p(\mathbb{S}^{d-1}) = \{f : \mathbb{S}^{d-1} \to \mathbb{R}| \int_{\mathbb{S}^{d-1}} f(\mathbf{u})u_d(\mathbf{u})\mathrm{d}u < \infty\}$. Since the $L^p$-convergence entails the point-wise convergence (Khoshnevisan, 2007), there exists a subsequence $\{\mu_{s(t(k))}\}_k$ such that $\lim_{k\to\infty} \mathrm{D}(\mathcal{R}_{\mathbf{u}}\mu_{s(t(k))} * \mathcal{N}_\sigma, \mathcal{R}_{\mathbf{u}}\mu * \mathcal{N}_\sigma) = 0$ almost everywhere for all $\mathbf{u} \in \mathbb{S}^{d-1}$. Recall that the divergence D metrizes the weak convergence in $\mathcal{P}(\mathbb{R})$ then $\mathcal{R}_{\mathbf{u}}\mu_{s(t(k))} * \mathcal{N}_\sigma \Rightarrow \mathcal{R}_{\mathbf{u}}\mu * \mathcal{N}_\sigma$ almost everywhere for all $\mathbf{u} \in \mathbb{S}^{d-1}$. Therefore, $\mathcal{R}_{\mathbf{u}}\mu_{s(t(k))} \Rightarrow \mathcal{R}_{\mathbf{u}}\mu$ almost everywhere for all $\mathbf{u} \in \mathbb{S}^{d-1}$. Using Cramér-Wold device (Huber, 2011), we get $\mu_{s(t(k))} \Rightarrow \mu$. Since the Lévy-Prokhorov distance metrizes the weak convergence, it entails that $\lim_{k\to\infty} \Lambda(\mu_{s(t(k))}, \mu_k) = 0$, that contradicts the fact that $\Lambda(\mu_{s(k)}, \mu) > \varepsilon$. We then conclude by contrapositive that $\mu_k \Rightarrow \mu$.

### A.3 Proof of Proposition 3.3: $\mathrm{G}_\sigma \mathrm{SD}_p$ is lower semi-continuous

Recall that the base divergence D is lower semi-continuous w.r.t. the weak topology in $\mathcal{P}(\mathbb{R})$, namely for every sequence of measures $\{\mu'_k\}_{k\in\mathbb{N}}$ and $\{\nu'_k\}_{k\in\mathbb{N}}$ in $\mathcal{P}(\mathbb{R})$ such that $\mu'_k \Rightarrow \mu'$ and $\nu'_k \Rightarrow \nu'$, one has $\mathrm{D}(\mu', \nu') \leq \liminf_{k\to\infty} \mathrm{D}(\mu'_k, \nu'_k)$.

Now, let $\{\mu_k\}_{k\in\mathbb{N}}$ and $\{\nu_k\}_{k\in\mathbb{N}}$ are two sequences of measure in $\mathcal{P}_p(\mathbb{R}^d)$ such that $\mu_k \Rightarrow \mu$ and $\nu_k \Rightarrow \nu$. By continuous mapping theorem (Bowers & Kalton, 2014) and Levy's continuity theorem, we obtain $\mathcal{R}_{\mathbf{u}}\mu_k * \mathcal{N}_\sigma \Rightarrow \mathcal{R}_{\mathbf{u}}\mu * \mathcal{N}_\sigma$ and $\mathcal{R}_{\mathbf{u}}\nu_k * \mathcal{N}_\sigma \Rightarrow \mathcal{R}_{\mathbf{u}}\nu * \mathcal{N}_\sigma$ for all $\mathbf{u} \in \mathbb{S}^{d-1}$. Since the base divergence D is a lower semi-continuous with respect to weak topology in $\mathcal{P}(\mathbb{R})$, then

$$\mathrm{D}^p(\mathcal{R}_{\mathbf{u}}\mu * \mathcal{N}_\sigma, \mathcal{R}_{\mathbf{u}}\nu * \mathcal{N}_\sigma) \leq \big( \liminf_{k\to\infty} \mathrm{D}(\mathcal{R}_{\mathbf{u}}\mu_k * \mathcal{N}_\sigma, \mathcal{R}_{\mathbf{u}}\nu_k * \mathcal{N}_\sigma) \big)^p \leq \liminf_{k\to\infty} \mathrm{D}^p(\mathcal{R}_{\mathbf{u}}\mu_k * \mathcal{N}_\sigma, \mathcal{R}_{\mathbf{u}}\nu_k * \mathcal{N}_\sigma).$$

It gives

$$\mathrm{G}_\sigma \mathrm{SD}_p(\mu, \nu) \leq \Big( \int_{\mathbb{S}^{d-1}} \liminf_{k\to\infty} \mathrm{D}^p(\mathcal{R}_{\mathbf{u}}\mu_k * \mathcal{N}_\sigma, \mathcal{R}_{\mathbf{u}}\nu_k * \mathcal{N}_\sigma) u_d(\mathbf{u})\mathrm{d}\mathbf{u} \Big)^{1/p}.$$

Furthermore, by application of Fatou's lemma (Bowers & Kalton, 2014), we get

$$\mathrm{G}_\sigma \mathrm{SD}_p(\mu, \nu) \leq \liminf_{k\to\infty} \Big( \int_{\mathbb{S}^{d-1}} \mathrm{D}^p(\mathcal{R}_{\mathbf{u}}\mu_k * \mathcal{N}_\sigma, \mathcal{R}_{\mathbf{u}}\nu_k * \mathcal{N}_\sigma) u_d(\mathbf{u})\mathrm{d}\mathbf{u} \Big)^{1/p} = \liminf_{k\to\infty} \mathrm{G}_\sigma \mathrm{SD}_p(\mu_k, \nu_k),$$

which is the desired result.

### A.4 Proofs of statistical properties

### A.4.1 Proof of Lemma 3.5: $\mathcal{R}_\mathbf{u}\hat{\mu}_n * \mathcal{N}_\sigma$ is an average of Gaussian mixture

Straighforwardly, for every Borelian $I \in \mathcal{B}(\mathbb{R})$, we have

$$
\begin{aligned}
\mathcal{R}_\mathbf{u}\hat{\mu}_n * \mathcal{N}_\sigma(I) &= \int_r \int_s \mathbf{1}_I(r+s)\mathrm{d}\{\frac{1}{n}\sum_{i=1}^n \delta_{\mathbf{u}^\top X_i}\}(r)\mathrm{d}\mathcal{N}_\sigma(s) \\
&= \frac{1}{n}\sum_{i=1}^n \int_s \mathbf{1}_I(\mathbf{u}^\top X_i + s)f_{\mathcal{N}_\sigma}(s)\mathrm{d}s \\
&= \frac{1}{n}\sum_{i=1}^n \int_{s'} \mathbf{1}_I(s')f_{\mathcal{N}_\sigma}(s' - \mathbf{u}^\top X_i)\mathrm{d}s' \\
&= \frac{1}{n}\sum_{i=1}^n \int_{s'} \mathbf{1}_I(s')f_{\mathcal{N}(\mathbf{u}^\top X_i, \sigma^2)}(s')\mathrm{d}s' \quad (\text{since } f_{\mathcal{N}_\sigma}(s' - \mathbf{u}^\top X_i) = f_{\mathcal{N}(\mathbf{u}^\top X_i, \sigma^2)}(s')) \\
&= \frac{1}{n}\sum_{i=1}^n \mathcal{N}(\mathbf{u}^\top X_i, \sigma^2)(I).
\end{aligned}
$$

Thanks to Theorem of Cramér and Wold (Cramér & Wold, 1936), we conclude the equality between the measures $\mathcal{R}_\mathbf{u}\hat{\mu}_n * \mathcal{N}_\sigma = \frac{1}{n}\sum_{i=1}^n \mathcal{N}(\mathbf{u}^\top X_i, \sigma^2)$.

### A.4.2 Proof of Proposition 3.8

Let us give first the overall structure of the proof. We we use frequently the triangle inequality for Wasserstein distances between the quantities $\hat{\mu}_n$, $\frac{1}{n}\mathcal{N}_\sigma(u^\top X_i, \sigma^2)$ and $\mathcal{R}_\mathbf{u}\mu * \mathcal{N}_\sigma$. We then obtain two quantities, **I** and **II** (see below for explicit), bounding $\mathbf{E}_{\mu^{\otimes n}|\mathcal{N}_\sigma^{\otimes n}}[\hat{\mathrm{G}}_\sigma \mathrm{SW}_p(\hat{\mu}_n, \mu)]$. To control **I** bound, we use a well known converging bound in Fournier & Guillin (2015) of Wasserstein distance between empirical and true measure. For **II** bound, we consider maximal TV-coupling in Villani (2009)] and use result of the $2p$-moment of absolute Gaussian random variable founded in Winkelbauer (2014).

On one hand, using triangle inequality of Wasserstein distance, we have

$$
\begin{aligned}
\mathbf{E}_{\mu^{\otimes n}|\mathcal{N}_\sigma^{\otimes n}}[\hat{\mathrm{G}}_\sigma \mathrm{SW}_p(\hat{\mu}_n, \mu)] &= \mathbf{E}_{\mu^{\otimes n}|\mathcal{N}_\sigma^{\otimes n}}\left[\left(\int_{\mathbb{S}^{d-1}} \mathrm{W}_p^p(\hat{\hat{\mu}}_n, R_\mathbf{u}\mu * \mathcal{N}_\sigma)u_d(\mathbf{u})\mathrm{d}\mathbf{u}\right)^{1/p}\right] \\
&\leq \left(\mathbf{E}_{\mu^{\otimes n}|\mathcal{N}_\sigma^{\otimes n}}\left[\int_{\mathbb{S}^{d-1}} \mathrm{W}_p^p(\hat{\hat{\mu}}_n, R_\mathbf{u}\mu * \mathcal{N}_\sigma)u_d(\mathbf{u})\mathrm{d}\mathbf{u}\right]\right)^{1/p} \\
&\leq \left(\int_{\mathbb{S}^{d-1}} \mathbf{E}_{\mu^{\otimes n}|\mathcal{N}_\sigma^{\otimes n}}[\mathrm{W}_p^p(\hat{\hat{\mu}}_n, R_\mathbf{u}\mu * \mathcal{N}_\sigma)]u_d(\mathbf{u})\mathrm{d}\mathbf{u}\right)^{1/p} \\
&\leq (\mathbf{I} + \mathbf{II})^{1/p}
\end{aligned}
$$

where

$$
\mathbf{I} \triangleq 2^{p-1}\int_{\mathbb{S}^{d-1}} \mathbf{E}_{\mu^{\otimes n}|\mathcal{N}_\sigma^{\otimes n}}\left[\mathrm{W}_p^p\left(\hat{\hat{\mu}}_n, \frac{1}{n}\sum_{i=1}^n \mathcal{N}(\mathbf{u}^\top X_i, \sigma^2)\right)\right]u_d(\mathbf{u})\mathrm{d}\mathbf{u}
$$

and

$$
\mathbf{II} \triangleq 2^{p-1}\int_{\mathbb{S}^{d-1}} \mathbf{E}_{\mu^{\otimes n}|\mathcal{N}_\sigma^{\otimes n}}\left[\mathrm{W}_p^p\left(\frac{1}{n}\sum_{i=1}^n \mathcal{N}(\mathbf{u}^\top X_i, \sigma^2), R_\mathbf{u}\mu * \mathcal{N}_\sigma)\right)\right]u_d(\mathbf{u})\mathrm{d}\mathbf{u}
$$

The proof is based on two steps to control the quantities **I** and **II**.
*Step 1: Control of **I**.*
Let us state the following lemma:

**Lemma A.1** (See proof of Theorem 1 in Fournier & Guillin (2015)). *Let $\eta \in \mathcal{P}(\mathbb{R})$ and let $p \geq 1$. Assume that $M_q(\eta) < \infty$ for some $q > p$. There exists a constant $C_{p,q}$ depending only on $p,q$ such that, for all $n \geq 1$,*

$$\mathbf{E}[\mathrm{W}_p^p(\hat{\eta}_n, \eta)] \leq C_{p,q} M_q(\eta)^{p/q} \Delta_n(p,q),$$

*where*

$$\Delta_n(p,q) = \begin{cases} n^{-1/2}\mathbf{1}_{q>2p}, \\ n^{-1/2}\log(n)\mathbf{1}_{q=2p} \\ n^{-(q-p)/q}\mathbf{1}_{p<q<2p}. \end{cases}.$$

We note that $\hat{\mu}_n$ is an empirical version of the Gausian mixture $\frac{1}{n}\sum_{i=1}^n \mathcal{N}_\sigma(u^\top X_i, \sigma^2)$. Then, by application of Lemma A.1, we get

$$\mathbf{E}_{\mu^{\otimes n}|\mathcal{N}_\sigma^{\otimes n}}\big[\mathrm{W}_p^p\big(\hat{\mu}_n, \frac{1}{n}\sum_{i=1}^n \mathcal{N}(\mathbf{u}^\top X_i, \sigma^2)\big)\big] \leq C_{p,q}\mathbf{E}_{\mu^{\otimes n}}\big[M_q^{p/q}\big(\frac{1}{n}\sum_{i=1}^n \mathcal{N}(\mathbf{u}^\top X_i, \sigma^2)\big)\big]\Delta_n(p,q).$$

Let us first upper bound the $q$-th moment of $M_q\big(\frac{1}{n}\sum_{i=1}^n \mathcal{N}(\mathbf{u}^\top X_i, \sigma^2)\big)$, for all $q \geq 1$. For all $\mathbf{u} \in \mathbb{S}^{d-1}$, we have

$$M_q\big(\frac{1}{n}\sum_{i=1}^n \mathcal{N}(\mathbf{u}^\top X_i, \sigma^2)\big) = \int_{\mathbb{R}} |t|^q \mathrm{d}(\frac{1}{n}\sum_{i=1}^n \mathcal{N}(\mathbf{u}^\top X_i, \sigma^2))(t) = \frac{1}{n}\sum_{i=1}^n M_q(|Z_{i,\mathbf{u}}|^q),$$

where $Z_{i,\mathbf{u}} \sim \mathcal{N}(\mathbf{u}^\top X_i, \sigma^2))$. By Equation (17) in Winkelbauer (2014) we have

$$M_q\big(\frac{1}{n}\sum_{i=1}^n \mathcal{N}(\mathbf{u}^\top X_i, \sigma^2)\big) = \frac{1}{n}\frac{2^{q/2}\sigma^q}{\sqrt{\pi}}\Gamma(\frac{q+1}{2})\sum_{i=1}^n {}_1F_1\big(-\frac{q}{2}, \frac{1}{2}; \frac{-(\mathbf{u}^\top X_i)^2}{2\sigma^2}\big).$$

Since $X_1, \ldots, X_n$ are i.i.d samples from $\mu$, it yields

$$\mathbf{E}_{\mu^{\otimes n}}\big[M_q^{p/q}\big(\frac{1}{n}\sum_{i=1}^n \mathcal{N}(\mathbf{u}^\top X_i, \sigma^2)\big)\big] = \frac{2^{q/2}\sigma^q}{\sqrt{\pi}}\Gamma(\frac{q+1}{2})\mathbf{E}_\mu\big[{}_1F_1\big(-\frac{q}{2}, \frac{1}{2}; \frac{-(\mathbf{u}^\top X)^2}{2\sigma^2}\big)\big] \quad (X \sim \mu)$$

$$= \frac{2^{q/2}\sigma^q}{\sqrt{\pi}}\Gamma(\frac{q+1}{2})\sum_{k=0}^\infty \frac{(-\frac{q}{2})_k}{(\frac{1}{2})_k}\frac{(-1)^k}{(2\sigma^2)^k k!}\mathbf{E}_\mu[(\mathbf{u}^\top X)^{2k}]$$

$$\leq \frac{2^{q/2}\sigma^q}{\sqrt{\pi}}\Gamma(\frac{q+1}{2})\sum_{k=0}^\infty \frac{(-\frac{q}{2})_k}{(\frac{1}{2})_k}\frac{(-1)^k}{(2\sigma^2)^k k!}M_{2k}(\mu).$$

Setting $q = 2p$ we have $\Delta_n(p,q) = \frac{\log n}{n}$, then

$$\mathbf{I} \leq 2^{2p-1}C_p\frac{\sigma^{2p}}{\sqrt{\pi}}\Gamma(\frac{2p+1}{2})\sum_{k=0}^\infty \frac{(-p)_k}{(\frac{1}{2})_k}\frac{(-1)^k}{(2\sigma^2)^k k!}M_{2k}(\mu)\frac{\log(n)}{n}.$$

*Step 2: Control of* **II**.

We follow the lines of proofs of Proposition 1 in Goldfeld et al. (2020) and Theorem 2 in Nietert et al. (2021). Using a coupling $\hat{\mu}_n$ and $\mathcal{R}_\mathbf{u}\mu * \mathcal{N}_\sigma)$ via the maximal TV-coupling (see Theorem 6.15 in Villani (2009)]), the control of the total variation of the Wasserstein distance, we get for any fixed $\mathbf{u} \in \mathbb{S}^{d-1}$

$$\mathrm{W}_p^p\big(\frac{1}{n}\sum_{i=1}^n \mathcal{N}(\mathbf{u}^\top X_i, \sigma^2), R_\mathbf{u}\mu * \mathcal{N}_\sigma)\big) \leq 2^{p-1}\int_{\mathbb{R}} |t|^p |h_{n,\mathbf{u}}(t) - g_\mathbf{u}(t)|dt,$$

where $h_{n,\mathbf{u}}$ and $g_{\mathbf{u}}$ are the densities associated with $\mu_n$ and $\mathcal{R}_{\mathbf{u}}\mu * \mathcal{N}_\sigma$, respectively. Let $f_{\sigma,\vartheta}$ the probability density function of $\mathcal{N}_{\sigma,\vartheta}$, i.e, $f_{\sigma,\vartheta}(t) = \frac{1}{\sqrt{2\pi(\sigma\vartheta)^2}}e^{-\frac{t^2}{2(\sigma\vartheta)^2}}$ for $\vartheta > 0$ to be specified later. An application of Cauchy-Schwarz inequality gives

$$
\mathbf{E}_{\mu^{\otimes n}|\mathcal{N}_\sigma^{\otimes n}}\Big[\,\mathrm{W}_p^p\big(\frac{1}{n}\sum_{i=1}^{n}\mathcal{N}(\mathbf{u}^\top X_i, \sigma^2), R_{\mathbf{u}}\mu * \mathcal{N}_\sigma)\big)\Big]
$$

$$
\leq 2^{p-1}\mathbf{E}_{\mu^{\otimes n}|\mathcal{N}_\sigma^{\otimes n}}\int_{\mathbb{R}}|t|^p\sqrt{f_{\sigma,\vartheta}(t)}\frac{|h_{n,\mathbf{u}}(t)-g_{\mathbf{u}}(t)|}{\sqrt{f_{\sigma,\vartheta}(t)}}dt
$$

$$
\leq 2^{p-1}\mathbf{E}_{\mu^{\otimes n}|\mathcal{N}_\sigma^{\otimes n}}\Big(\int_{\mathbb{R}}|t|^{2p}f_{\sigma,\vartheta}(t)dt\Big)^{\frac{1}{2}}\Big(\int_{\mathbb{R}}\frac{(h_{n,\mathbf{u}}(t)-g_{\mathbf{u}}(t))^2}{f_{\sigma,\vartheta}(t)}dt\Big)^{\frac{1}{2}}
$$

$$
\leq 2^{p-1}\Big(\int_{\mathbb{R}}|t|^{2p}f_{\sigma,\vartheta}(t)dt\Big)^{\frac{1}{2}}\Big(\int_{\mathbb{R}}\mathbf{E}_{\mu^{\otimes n}|\mathcal{N}_\sigma^{\otimes n}}\frac{(h_{n,\mathbf{u}}(t)-g_{\mathbf{u}}(t))^2}{f_{\sigma,\vartheta}(t)}dt\Big)^{\frac{1}{2}}.
$$

Note that $\int_{\mathbb{R}}|t|^{2p}f_{\sigma,\vartheta}(t)dt$ is the $2p$-th moment of $|\mathcal{N}_{\sigma,\vartheta}(t)|$ equals to (see Equation (18) in Winkelbauer (2014))

$$
\int_{\mathbb{R}}|t|^{2p}f_{\sigma,\vartheta}(t)dt = \frac{(\sigma\vartheta)^{2p}2^p}{\sqrt{\pi}}\Gamma\big(\frac{2p+1}{2}\big).
$$

Moreover,

$$
h_{n,\mathbf{u}}(t) = \frac{1}{n}\sum_{i=1}^{n}\mathrm{d}\mathcal{N}(\mathbf{u}^\top X_i, \sigma^2)(t) = \frac{1}{n}\sum_{i=1}^{n}f_{\sigma,\vartheta}(t-\mathbf{u}^\top X_i),
$$

It is clear to see that $h_{n,\mathbf{u}}(t)$ is a sum of i.i.d. terms with expectation $g_{\mathbf{u}}(t)$, which implies

$$
\mathbf{E}_{\mu^{\otimes n}|\mathcal{N}_\sigma^{\otimes n}}\big[(h_{n,\mathbf{u}}(t)-g_{\mathbf{u}}(t))^2\big] = \mathbf{V}_{\mu^{\otimes n}}\Big[\frac{1}{n}\sum_{i=1}^{n}f_{\sigma,\vartheta}(t-\mathbf{u}^\top X_i)\Big]
$$

$$
= \frac{1}{n}\mathbf{V}_\mu[f_{\sigma,\vartheta}(t-\mathbf{u}^\top X]
$$

$$
\leq \frac{1}{n}\mathbf{E}_\mu[(f_{\sigma,\vartheta}(t-\mathbf{u}^\top X)^2]
$$

$$
\leq \frac{(2\pi\sigma^2)^{-1}}{n}\mathbf{E}_\mu[e^{\frac{-1}{\sigma^2}(t-\mathbf{u}^\top X)^2}].
$$

Now

$$
\mathbf{E}_\mu[e^{\frac{-(t-\mathbf{u}^\top X)^2}{\sigma^2}}] = \int_{\|x\|\leq\frac{|t|}{2}}e^{\frac{-1}{\sigma^2}(t-\mathbf{u}^\top x)^2}\mathrm{d}\mu(x) + \int_{\|x\|>\frac{|t|}{2}}e^{\frac{-1}{\sigma^2}(t-\mathbf{u}^\top x)^2}\mathrm{d}\mu(x).
$$

Remark that when $\|x\| \leq \frac{|t|}{2}$, then $(t-\mathbf{u}^\top X)^2 \geq |t|^2 - |\mathbf{u}^\top x|^2 \geq |t|^2 - \|x\|^2$ (since $\|u\|^2 = 1$). We get $(t-\mathbf{u}^\top X)^2 \geq \frac{|t|^2}{4}$ and hence

$$
\int_{\|x\|\leq\frac{|t|}{2}}e^{\frac{-1}{\sigma^2}(t-\mathbf{u}^\top x)^2}\mathrm{d}\mu(x) \leq e^{\frac{-t^2}{4\sigma^2}} \text{ and } \int_{\|x\|>\frac{|t|}{2}}e^{\frac{-1}{\sigma^2}(t-\mathbf{u}^\top x)^2}\mathrm{d}\mu(x) \leq \mathbf{P}\big[\|X\| > \frac{|t|}{2}\big]
$$

This gives,

$$
\int_{\mathbb{R}}\mathbf{E}_{\mu^{\otimes n}|\mathcal{N}_\sigma^{\otimes n}}\frac{(h_{n,\mathbf{u}}(t)-g_{\mathbf{u}}(t))^2}{f_{\sigma,\vartheta}(t)}\mathrm{d}t \leq \frac{(2\pi\sigma^2)^{-1}(\sqrt{2\pi}\sigma\vartheta)}{n}\Big(\int_{\mathbb{R}}e^{\frac{t^2}{2(\sigma\vartheta)^2}}e^{\frac{-t^2}{4\sigma^2}}\mathrm{d}t + \int_{\mathbb{R}}e^{\frac{t^2}{2(\sigma\vartheta)^2}}\mathbf{P}\big[\|X\| > \frac{|t|}{2}\big]\mathrm{d}t\Big).
$$

Note that the integral $\int_{\mathbb{R}} e^{\frac{t^2}{2(\sigma\vartheta)^2}} e^{\frac{-t^2}{4\sigma^2}} \mathrm{d}t = \int_{\mathbb{R}} e^{-\left(\frac{1}{2}-\frac{1}{\vartheta^2}\right)\frac{t^2}{2\sigma^2}} \mathrm{d}t$ is finite if and only if $\frac{1}{2} - \frac{1}{\vartheta^2} > 0$ namely $\vartheta > \sqrt{2}$ and its value is given by

$$\int_{\mathbb{R}} e^{\frac{t^2}{2(\sigma\vartheta)^2}} e^{\frac{-t^2}{4\sigma^2}} \mathrm{d}t = \sqrt{\frac{2\pi\sigma^2}{\frac{1}{2}-\frac{1}{\vartheta^2}}} = \sqrt{\frac{4\pi\sigma^2\vartheta^2}{\vartheta^2-2}}.$$

For the second integral

$$\int_{\mathbb{R}} e^{\frac{t^2}{2(\sigma\vartheta)^2}} \mathbf{P}\left[\|X\| > \frac{|t|}{2}\right] \mathrm{d}t = 2\int_0^\infty e^{\frac{t^2}{2(\sigma\vartheta)^2}} \mathbf{P}\left[\|X\| > \frac{t}{2}\right] \mathrm{d}t = 4\int_0^\infty e^{\frac{2\xi^2}{\sigma^2\vartheta^2}} \mathbf{P}\left[\|X\| > \xi\right] \mathrm{d}\xi$$

Then,

$$\mathbf{II} \le n^{-1/2} 4^{p-1} \left\{ (2\pi\sigma^2)^{-1}(\sqrt{2\pi}\sigma\vartheta)\frac{(\sigma\vartheta)^{2p}2^p}{\sqrt{\pi}}\Gamma\left(\frac{2p+1}{2}\right) \right\}^{\frac{1}{2}} \left( \sqrt{\frac{4\pi\sigma^2\vartheta^2}{\vartheta^2-2}} + 4\int_0^\infty e^{\frac{2\xi^2}{\sigma^2\vartheta^2}} \mathbf{P}\left[\|X\| > \xi\right] \mathrm{d}\xi \right)^{\frac{1}{2}}.$$

this gives the desired result using the fact that $(a+b)^{1/p} \le a^{1/p} + b^{1/p}$, for $a, b \ge 0$.

### A.4.3 Proof of Proposition 3.11

Using triangle inequality, we have

$$\mathrm{W}_p(\hat{\hat{\mu}}_n, \hat{\nu}_n) \le \mathrm{W}_p(\hat{\hat{\mu}}_n, \mathcal{R}_{\mathbf{u}}\mu * \mathcal{N}_\sigma) + \mathrm{W}_p(\mathcal{R}_{\mathbf{u}}\mu * \mathcal{N}_\sigma, \mathcal{R}_{\mathbf{u}}\nu * \mathcal{N}_\sigma) + \mathrm{W}_p(\mathcal{R}_{\mathbf{u}}\nu * \mathcal{N}_\sigma, \hat{\nu}_n).$$

and then

$$\mathrm{W}_p^p(\hat{\hat{\mu}}_n, \hat{\nu}_n) \le 3^{p-1}\left\{ \mathrm{W}_p^p(\hat{\hat{\mu}}_n, \mathcal{R}_{\mathbf{u}}\mu * \mathcal{N}_\sigma) + \mathrm{W}_p^p(\mathcal{R}_{\mathbf{u}}\mu * \mathcal{N}_\sigma, \mathcal{R}_{\mathbf{u}}\nu * \mathcal{N}_\sigma) + \mathrm{W}_p^p(\mathcal{R}_{\mathbf{u}}\nu * \mathcal{N}_\sigma, \hat{\nu}_n) \right\}.$$

This implies that

$$\begin{aligned}
\mathbf{E}_{\mu^{\otimes n}|\mathcal{N}_\sigma^{\otimes n}} & \mathbf{E}_{\nu^{\otimes n}|\mathcal{N}_\sigma^{\otimes n}} [\hat{\mathrm{G}}_\sigma \mathrm{SW}_p(\hat{\mu}_n, \hat{\nu}_n)] \\
& \le 3^{1-\frac{1}{p}} \mathrm{G}_\sigma \mathrm{SW}_p(\mu, \nu) + 3^{1-\frac{1}{p}} \mathbf{E}_{\mu^{\otimes n}|\mathcal{N}_\sigma^{\otimes n}}[\hat{\mathrm{G}}_\sigma \mathrm{SW}_p(\hat{\mu}_n, \mu)] + 3^{1-\frac{1}{p}} \mathbf{E}_{\nu^{\otimes n}|\mathcal{N}_\sigma^{\otimes n}}[\hat{\mathrm{G}}_\sigma \mathrm{SW}_p(\hat{\nu}_n, \nu)].
\end{aligned}$$

By application of Proposition 3.8, it yields This gives that

$$\mathbf{E}_{\mu^{\otimes n}|\mathcal{N}_\sigma^{\otimes n}} \mathbf{E}_{\nu^{\otimes n}|\mathcal{N}_\sigma^{\otimes n}} [\hat{\mathrm{G}}_\sigma \mathrm{SW}_p(\hat{\mu}_n, \hat{\nu}_n)] \le 3^{1-\frac{1}{p}} \mathrm{G}_\sigma \mathrm{SW}_p(\mu, \nu) + 3\Xi_{p,\sigma,\vartheta}\frac{1}{n^{1/2p}} + 3^{1-\frac{1}{p}}(\Upsilon_{p,\sigma,\mu} + \Upsilon_{p,\sigma,\nu})\frac{(\log n)^{1/p}}{n^{1/p}}$$

This ends the proof of the first statement in Proposition 3.11. For the second one, we also use a triangle inequality

$$\mathrm{W}_p^p(\mathcal{R}_{\mathbf{u}}\mu * \mathcal{N}_\sigma, \mathcal{R}_{\mathbf{u}}\nu * \mathcal{N}_\sigma) \le 3^{p-1}\left\{ \mathrm{W}_p^p(\mathcal{R}_{\mathbf{u}}\mu * \mathcal{N}_\sigma, \hat{\hat{\mu}}_n) + \mathrm{W}_p^p(\hat{\hat{\mu}}_n, \hat{\nu}_n) + \mathrm{W}_p^p(\hat{\nu}_n), \mathcal{R}_{\mathbf{u}}\nu * \mathcal{N}_\sigma \right\}.$$

Then we control each term as we did before.

### A.5 Proof of Proposition 3.12: projection complexity

Using Holder's inequality, we have

$$\begin{aligned}
\mathbf{E}_{u_d^{\otimes L}}\left[\left|\widehat{\mathrm{G}_\sigma \mathrm{SD}_p}^p(\mu, \nu) - \mathrm{G}_\sigma \mathrm{SD}_p^p(\mu, \nu)\right|\right] & \le \left(\mathbf{E}_{u_d^{\otimes L}}\left[\left[\left|\widehat{\mathrm{G}_\sigma \mathrm{SD}_p}^p(\mu, \nu) - \mathrm{G}_\sigma \mathrm{SD}_p^p(\mu, \nu)\right|^2\right]\right]\right)^{1/2} \\
& = \left(\mathbf{V}_{u_d^{\otimes L}}\left[\left[\widehat{\mathrm{G}_\sigma \mathrm{SD}_p}^p(\mu, \nu)\right]\right]\right)^{1/2} \\
& = \frac{A(p, \sigma)}{L^{1/2}}.
\end{aligned}$$

### A.6 Proof of Corollary 3.13: overall complexity ($p = 1$)

By application of triangle inequality, one has

$$|\widehat{\hat{G}_\sigma SW}(\hat{\mu}_n, \hat{\nu}_n) - G_\sigma SW(\mu, \nu)| \leq |\widehat{\hat{G}_\sigma SW}(\hat{\mu}_n, \hat{\nu}_n) - \hat{G}_\sigma SW(\hat{\mu}_n, \hat{\nu}_n)| + |\hat{G}_\sigma SW(\hat{\mu}_n, \hat{\nu}_n) - G_\sigma SW(\mu, \nu)|$$

Using Proposition 3.12, we have

$$\mathbf{E}_{u_d^{\otimes L}}\left[|\widehat{\hat{G}_\sigma SW}(\hat{\mu}_n, \hat{\nu}_n) - \hat{G}_\sigma SW(\hat{\mu}_n, \hat{\nu}_n)|\right] \leq \frac{\hat{A}_\sigma}{\sqrt{L}} := \frac{\{\mathbf{V}_{\mathbf{u}\sim u_d}[W(\hat{\mu}_n, \hat{\nu}_n)]\}^{1/2}}{\sqrt{L}}.$$

Using Proposition 3.11 for $p = 1$ we get,

$$\mathbf{E}_{\mu^{\otimes n}|\mathcal{N}_\sigma^{\otimes n}}\mathbf{E}_{\nu^{\otimes n}|\mathcal{N}_\sigma^{\otimes n}}[|\hat{G}_\sigma SW(\hat{\mu}_n, \hat{\nu}_n) - G_\sigma SW(\mu, \nu)|] \leq 3\Xi_{1,\sigma,\vartheta}\frac{1}{\sqrt{n}} + (\Upsilon_{1,\sigma,\mu} + \Upsilon_{1,\sigma,\nu})\frac{\log n}{n}.$$

Therefore, by applying the expectations with respect to the projection and sampling we obtain

$$\mathbf{E}_{u_d^{\otimes L}}\mathbf{E}_{\mu^{\otimes n}|\mathcal{N}_\sigma^{\otimes n}}\mathbf{E}_{\nu^{\otimes n}|\mathcal{N}_\sigma^{\otimes n}}\left[|\widehat{\hat{G}_\sigma SW}(\hat{\mu}_n, \hat{\nu}_n) - G_\sigma SW(\mu, \nu)|\right]$$
$$\leq \frac{1}{\sqrt{L}}\mathbf{E}_{\mu^{\otimes n}|\mathcal{N}_\sigma^{\otimes n}}\mathbf{E}_{\nu^{\otimes n}|\mathcal{N}_\sigma^{\otimes n}}[\hat{A}_\sigma] + 3\Xi_{1,\sigma,\vartheta}\frac{1}{\sqrt{n}} + (\Upsilon_{1,\sigma,\mu} + \Upsilon_{1,\sigma,\nu})\frac{\log n}{n}.$$

By Jensen inequality, we have

$$\mathbf{E}_{\mu^{\otimes n}|\mathcal{N}_\sigma^{\otimes n}}\mathbf{E}_{\nu^{\otimes n}|\mathcal{N}_\sigma^{\otimes n}}[\hat{A}_\sigma] \leq \left\{\mathbf{E}_{\mu^{\otimes n}|\mathcal{N}_\sigma^{\otimes n}}\mathbf{E}_{\nu^{\otimes n}|\mathcal{N}_\sigma^{\otimes n}}[\mathbf{V}_{\mathbf{u}\sim u_d}[W(\hat{\mu}_n, \hat{\nu}_n)]]\right\}^{1/2}.$$

### A.7 Proof of Proposition 3.14

For all $\mathbf{u} \in \mathbb{S}^{d-1}$ we have $\mathcal{R}_{\mathbf{u}}\mu, \mathcal{R}_{\mathbf{u}}\nu \in \mathcal{P}(\mathbb{R})$. By application of the inequality of noise level satisfied by D in one dimension we get

$$D^p(\mathcal{R}_{\mathbf{u}}\mu * \mathcal{N}_{\sigma_2}, \mathcal{R}_{\mathbf{u}}\nu * \mathcal{N}_{\sigma_2}) \leq D^p(\mathcal{R}_{\mathbf{u}}\mu * \mathcal{N}_{\sigma_1}, \mathcal{R}_{\mathbf{u}}\nu * \mathcal{N}_{\sigma_1}).$$

Then, computing the expectation over the projections $\mathbf{u}$ since the divergence is non-negative concludes the proof.

### A.8 Proof of Proposition 3.16: relation between $G_\sigma SW^p(\mu, \nu)$ under two noise levels

First, using the contractive property of convolution (see Lemma 3 in Nietert et al. (2021)), stating that for any probability measure $\alpha \in \mathcal{P}(\mathbb{R})$, $W_p(\mu*\alpha, \nu*\alpha) \leq W_p(\mu, \nu)$. Hence $W_p^p(\mu*\mathcal{N}_{\sigma_2}, \nu*\mathcal{N}_{\sigma_2}) \leq W_p^p(\mu*\mathcal{N}_{\sigma_1}, \nu*\mathcal{N}_{\sigma_1})$. Now using Proposition 3.14 of the oreder relation satisfied by $G_\sigma SW^p$ yields

$$G_{\sigma_2}SW_p(\mu, \nu) \leq G_{\sigma_1}SW_p(\mu, \nu).$$

In the other direction, we have that $\mathcal{N}_{\sigma_2} = \mathcal{N}_{\sigma_1} * \mathcal{N}_{\sqrt{\sigma_2^2 - \sigma_1^2}}$ (similarly for $\mathcal{N}_{\sigma_1}$). Setting the following random variables: $X_{\mathbf{u}} \sim \mathcal{R}_{\mathbf{u}}\mu, Y_{\mathbf{u}} \sim \mathcal{R}_{\mathbf{u}}\nu, Z_X \sim \mathcal{N}_{\sigma_1}, Z_Y \sim \mathcal{N}_{\sigma_1}, Z_X' \sim \mathcal{N}_{\sqrt{\sigma_2^2 - \sigma_1^2}}, Z_Y' \sim \mathcal{N}_{\sqrt{\sigma_2^2 - \sigma_1^2}}$. The sliced Wasserstein distance $W_p^p(\mathcal{R}_{\mathbf{u}}\mu * \mathcal{N}_{\sigma_2}, \mathcal{R}_{\mathbf{u}}\nu * \mathcal{N}_{\sigma_2})$ is given as a minimization over couplings $(X_{\mathbf{u}}, Z_X, Z_X')$ and $(Y_{\mathbf{u}}, Z_Y, Z_Y')$, namely

$$W_p^p(\mathcal{R}_{\mathbf{u}}\mu * \mathcal{N}_{\sigma_2}, \mathcal{R}_{\mathbf{u}}\nu * \mathcal{N}_{\sigma_2}) = \inf_{\substack{X_{\mathbf{u}}, Z_X, Z_X' \\ Y_{\mathbf{u}}, Z_Y, Z_Y'}} \mathbf{E}\left[|((X_{\mathbf{u}} + Z_X) - (Y_{\mathbf{u}} + Z_Y)) + (Z_X' - Z_Y')|^p\right]$$

Using the inequality $\mathbf{E}[|U+V|^p] - 2^{p-1}\mathbf{E}[|W|^p] \leq 2^{p-1}\mathbf{E}[|U+V+W|^p]$ for any random variables $U, V, W \in \mathbb{L}_p$ integrable, we obtain,

$$2^{p-1}\mathbf{E}\left[|(X_{\mathbf{u}} + Z_X) - (Y_{\mathbf{u}} + Z_Y) + (Z_X' + Z_Y')|^p\right] \geq \mathbf{E}\left[|(X_{\mathbf{u}} + Z_X) - (Y_{\mathbf{u}} + Z_Y)|^p\right] - 2^{p-1}\mathbf{E}\left[|(Z_X' - Z_Y')|^p\right].$$

Hence,

$$2^{p-1}\mathrm{W}_p^p(\mathcal{R}_{\mathbf{u}}\mu * \mathcal{N}_{\sigma_2}, \mathcal{R}_{\mathbf{u}}\nu * \mathcal{N}_{\sigma_2}) \geq \inf\Big(\mathbf{E}\big[|(X_{\mathbf{u}} + Z_X) - (Y_{\mathbf{u}} + Z_Y)|^p\big] - 2^{p-1}\mathbf{E}\big[|(Z_X' - Z_Y')|^p\big]\Big)$$
$$\geq \mathrm{W}_p^p(\mathcal{R}_{\mathbf{u}}\mu * \mathcal{N}_{\sigma_1}, \mathcal{R}_{\mathbf{u}}\nu * \mathcal{N}_{\sigma_1}) - 2^{p-1}\sup\mathbf{E}\big[|(Z_X' - Z_Y')|^p\big]$$
$$\geq \mathrm{W}_p^p(\mathcal{R}_{\mathbf{u}}\mu * \mathcal{N}_{\sigma_1}, \mathcal{R}_{\mathbf{u}}\nu * \mathcal{N}_{\sigma_1}) - 2^{2p}\sup\mathbf{E}\big[|(Z_X')|^p\big].$$

Hence,

$$\mathrm{G}_{\sigma_1}\mathrm{SW}_p(\mu,\nu) \leq 2^{1-\frac{1}{p}}\,\mathrm{G}_{\sigma_2}\mathrm{SW}_p(\mu,\nu) + 4\big(\sup\mathbf{E}\big[|(Z_X')|^p\big]\big)^{1/p}.$$

Finally, for any $p \geq 1$ the $p$-th moment of $|\mathcal{N}_\sigma|$ satisfies $\mathbf{E}[|\mathcal{N}_\sigma|^p] = \frac{2^p\Gamma((p+1)/2)}{\Gamma(1/2)}\sigma^{2p} \leq 2^{p/2}\sigma^{2p}$, then

$$\mathrm{G}_{\sigma_1}\mathrm{SW}_p(\mu,\nu) \leq 2^{1-\frac{1}{p}}\,\mathrm{G}_{\sigma_2}\mathrm{SW}_p(\mu,\nu) + 2^{\frac{5}{2}}(\sigma_2^2 - \sigma_1^2),$$

and concludes the proof.

### A.9 Proof of Proposition 3.17: continuity of the smoothed Gaussian sliced Wasserstein w.r.t. $\sigma$

From Lemma 1 in (Nietert et al., 2021), we know that the Gaussian-smoothed Wasserstein is continuous with respect to $\sigma$, for any distribution $\mathcal{R}_{\mathbf{u}}\nu$ and $\mathcal{R}_{\mathbf{u}}\mu$. In addition, for any $\mathbf{u}$, we have $\mathrm{W}_p(\mathcal{R}_{\mathbf{u}}\nu * \mathcal{N}_\sigma, \mathcal{R}_{\mathbf{u}}\mu * \mathcal{N}_\sigma) \leq \mathrm{W}_p(\mathcal{R}_{\mathbf{u}}\nu, \mathcal{R}_{\mathbf{u}}\mu)$. Then by applying Lebesgue's dominated convergence theorem (Bowers & Kalton, 2014) to the above inequality with $\mathrm{W}_p(\mathcal{R}_{\mathbf{u}}\nu, \mathcal{R}_{\mathbf{u}}\mu)$ as a dominating function, that is $u_d$-almost everywhere integrable because both measures are in $\mathcal{P}_p(\mathbb{R}^d)$, we then conclude that the Gaussian-smoothed SWD is continuous w.r.t. $\sigma$.

### A.10 Proof of Proposition 3.18: continuity of the smoothed sliced squared-MMD w.r.t. $\sigma$

Let us first recall the definition of the MMD divergence. Let $k : \mathbb{R} \times \mathbb{R} \to \mathbb{R}$ be a measurable bounded kernel on $\mathbb{R}$ and consider the reproducing kernel Hilbert space (RKHS) $\mathcal{H}_k$ associated with $k$ and equipped with inner product $< \cdot, \cdot >_{\mathcal{H}_k}$ and norm $\|\cdot\|_{\mathcal{H}_k}$. Let $\mathcal{P}_{\mathcal{H}_k}(\mathbb{R})$ be the set of probability measures $\eta$ such that $\int_{\mathbb{R}} \sqrt{k(t,t)}d\eta(x) < \infty$. The kernel mean embedding is defined as $\Phi_k(\eta) = \int_{\mathbb{R}} k(\cdot, t)d\eta(t)$. The squared-maximum mean discrepancy between $\eta, \zeta \in \mathcal{P}(\mathbb{R})$ denoted as $\mathrm{MMD} : \mathcal{P}_{\mathcal{H}_k}(\mathbb{R}) \times \mathcal{P}_{\mathcal{H}_k}(\mathbb{R}) \to \mathbb{R}_+$ is expressed as the distance between two such kernel mean embeddings. It is defined as Gretton et al. (2012)

$$\mathrm{MMD}^2(\eta,\zeta) = \|\Phi_k(\eta) - \Phi_k(\zeta)\|_{\mathcal{H}_k}^2 = \mathbf{E}_{T,T'\sim\eta}[k(T,T')] - 2\mathbf{E}_{T\sim\eta,R\sim\zeta}[k(T,R)] + \mathbf{E}_{R,R'\sim\zeta}[k(R,R')]$$

where $T$ and $T'$ are independent random variables drawn according to $\eta$, $R$ and $R'$ are independent random variables drawn according to $\zeta$, and $T$ is independent of $R$. We define the Gaussian Smoothed Sliced squared-MMD as follows:

$$\mathrm{G}_\sigma\mathrm{MMD}^2(\mu,\nu) = \int_{\mathbb{S}^{d-1}} \|\Phi_k(\mathcal{R}_{\mathbf{u}}\mu * \mathcal{N}_\sigma) - \Phi_k(\mathcal{R}_{\mathbf{u}}\nu * \mathcal{N}_\sigma)\|_{\mathcal{H}_k}^2 u_d(\mathbf{u})\mathrm{d}\mathbf{u}$$
$$= \int_{\mathbb{S}^{d-1}} \big(\mathbf{E}_{T,T'\sim\mathcal{R}_{\mathbf{u}}\mu*\mathcal{N}_\sigma}[k(T,T')] - 2\mathbf{E}_{T\sim\mathcal{R}_{\mathbf{u}}\mu*\mathcal{N}_\sigma, R\sim\mathcal{R}_{\mathbf{u}}\nu*\mathcal{N}_\sigma}[k(T,R)]$$
$$+ \mathbf{E}_{R,R'\sim\mathcal{R}_{\mathbf{u}}\nu*\mathcal{N}_\sigma}[k(R,R')]\big)u_d(\mathbf{u})\mathrm{d}\mathbf{u}.$$

From the definition of the smoothed sliced squared-MMD, we have

$$\mathbf{E}_{T,T'\sim\mathcal{R}_{\mathbf{u}}\mu*\mathcal{N}_\sigma}[k(T,T')] = \iint_{\mathbb{R}\times\mathbb{R}} k(t,t')\mathrm{d}\mathcal{R}_{\mathbf{u}}\mu * \mathcal{N}_\sigma(t)\mathrm{d}\mathcal{R}_{\mathbf{u}}\mu * \mathcal{N}_\sigma(t')$$
$$= \iint_{\mathbb{R}\times\mathbb{R}} \Big(\int_{\mathbb{R}} k(t+z,t')\mathrm{d}\mathcal{R}_{\mathbf{u}}\mu(z)\mathcal{N}_\sigma(t)\Big)\mathrm{d}\mathcal{R}_{\mathbf{u}}\mu * \mathcal{N}_\sigma(t')$$
$$= \iint_{\mathbb{R}\times\mathbb{R}} \Big(\int_{\mathbb{R}^d} k(t+\mathbf{u}^\top x,t')\mathrm{d}\mu(x)\mathcal{N}_\sigma(t)\Big)\mathrm{d}\mathcal{R}_{\mathbf{u}}\mu * \mathcal{N}_\sigma(t')$$
$$= \iint_{\mathbb{R}\times\mathbb{R}} \iint_{\mathbb{R}^d\times\mathbb{R}^d} k(t+\mathbf{u}^\top x, t'+\mathbf{u}^\top x')\mathrm{d}\mu(x)\mathrm{d}\mu(x')\mathrm{d}\mathcal{N}_\sigma(t)\mathrm{d}\mathcal{N}_\sigma(t').$$

Similarly,

$$\mathbf{E}_{R,R'\sim\mathcal{R}_{\mathbf{u}}\nu*\mathcal{N}_\sigma}[k(R,R')] = \iint_{\mathbb{R}\times\mathbb{R}} \iint_{\mathbb{R}^d\times\mathbb{R}^d} k(r+\mathbf{u}^\top y, r'+\mathbf{u}^\top y')\mathrm{d}\nu(y)\mathrm{d}\nu(y')\mathrm{d}\mathcal{N}_\sigma(r)\mathrm{d}\mathcal{N}_\sigma(r')$$

and

$$\mathbf{E}_{T\sim\mathcal{R}_{\mathbf{u}}\mu*\mathcal{N}_\sigma, R\sim\mathcal{R}_{\mathbf{u}}\nu*\mathcal{N}_\sigma}[k(T,R)] = \iint_{\mathbb{R}\times\mathbb{R}} \iint_{\mathbb{R}^d\times\mathbb{R}^d} k(t+\mathbf{u}^\top x, r+\mathbf{u}^\top y)\mathrm{d}\mu(x)\mathrm{d}\nu(y)\mathrm{d}\mathcal{N}_\sigma(t)\mathrm{d}\mathcal{N}_\sigma(r).$$

Together the assumption of boundness of the kernel function $k$ and the continuity of integrals, the three latter terms are continuous functions w.r.t. $\sigma \in (0,\infty)$. Again by the boundness of the kernel function $k$, there exists a positive finite constant $C_k$ such that

$$\left|\mathbf{E}_{T,T'\sim\mathcal{R}_{\mathbf{u}}\mu*\mathcal{N}_\sigma}[k(T,T')] - 2\mathbf{E}_{T\sim\mathcal{R}_{\mathbf{u}}\mu*\mathcal{N}_\sigma, R\sim\mathcal{R}_{\mathbf{u}}\nu*\mathcal{N}_\sigma}[k(T,R)] + \mathbf{E}_{R,R'\sim\mathcal{R}_{\mathbf{u}}\nu*\mathcal{N}_\sigma}[k(R,R')]\right| \le 4C_k.$$

We conclude the continuity of $\sigma \mapsto \mathrm{G}_\sigma\mathrm{MMD}^2(\mu,\nu)$ by an application of the continuity of integrals.

## B   Additional experiments

### B.1   Sample complexity on CIFAR dataset

We have also evaluated the sample complexity for the CIFAR dataset by sampling sets of increasing size. Results reported in Figure 6 confirms the findings obtained from the toy dataset.

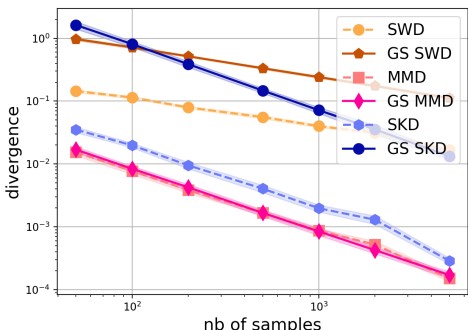

Figure 6: Measuring the divergence between two sets of samples drawn iid from the CIFAR10 dataset. We compare three sliced divergences and their Gaussian smoothed versions with a $\sigma = 3$.

### B.2   Identity of indiscernibles

The second experiment aims at checking whether our divergences converge towards a small value when the distributions to be compared are the same. For this, we consider samples from distributions $\mu$ and $\nu$ chosen as normal distributions with respectively mean $2 \times \mathbf{1}_d$ and $s\mathbf{1}_d$ with varying $s$ (noted as the displacement). Results are depicted in Figure 7. We can see that all methods are able to attain their minimum when $s = 2$. Interestingly, the gap between the Gaussian smoothed and non-smoothed divergences for Wasserstein and Sinkhorn is almost indiscernible as the distance between distribution increases.

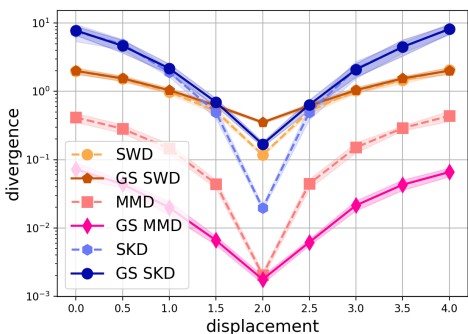

Figure 7: Measuring the divergence between two sets of samples in $\mathbb{R}^{50}$, one with mean $2\mathbf{1}_d$ and the other with mean $s\mathbf{1}_d$ with increasing $s$. We compare three sliced divergences and their Gaussian smoothed version with a $\sigma = 3$.

