# OpenReview forum: "Gaussian-Smoothed Sliced Probability Divergences"
_TMLR — Accepted by TMLR_

### Review · Reviewer_vY8A · 2024-06-03

**Summary Of Contributions:**

In this paper, the author provides the theoretical properties of the Gaussian smoothed sliced Wasserstein distance, including its topology and sample complexity. They also extend some of these theoretical results to other divergences. A series of experiments have been done to support these results.

**Audience:**

Yes

**Claims And Evidence:**

Yes

**Requested Changes:**

I suggest rewriting Proposition 3.16 as a stability result of $G_{\sigma}SW^p$ as that in [2].

**Strengths And Weaknesses:**

Strengths:

1. This paper is well-written and easy to follow.
2. They introduce the double empirical distribution and establish the $\mathcal{O} (n^{-1/2})$ sample complexity results of the smoothed sliced Wasserstein distance.

Questions:
1. In [1], they develop the limit distribution theory for smooth $p$ Wasserstein distance. Intuitively, would similar results hold for smoothed sliced $p$ Wasserstein distance?
2. In [2], they show the convergence of transport plans. Can you derive similar results?

[1] Goldfeld, Ziv, et al. "Limit distribution theory for smooth p-Wasserstein distances." The Annals of Applied Probability 34.2 (2024): 2447-2487.

[2] Nietert, Sloan, Ziv Goldfeld, and Kengo Kato. "Smooth $ p $-Wasserstein distance: structure, empirical approximation, and statistical applications." International Conference on Machine Learning. PMLR, 2021.

---

> ### Author Response · Authors · 2024-09-03
>
> We thank the Reviewer for his time, consideration, and invaluable feedback. Next, we provide point-by-point answers to the concerns pointed out by the Reviewer.
>
> **In [Goldfel et al., Ann. Proba. 2024], they develop the limit distribution theory for smooth $p$ Wasserstein distance. Intuitively, would similar results hold for smoothed sliced $p$ Wasserstein distance**
>
> Let us first recall the definition of the one sample double empirical smoothed Gaussian sliced Wasserstein
> \begin{equation}
> \hat{\text{G}}\_{\sigma}\text{SW}\_p (\hat \mu\_n,\mu) = \big(\int\_{\mathbb{S}^{d-1}} W\_p^p(\hat{\hat\mu}\_{n}, \mathcal{R}\_{\boldsymbol{\rm u}} \mu * \mathcal{N}\_{\sigma} )u\_d(\boldsymbol{\rm u})\rm{d}\boldsymbol{u}\Big)^{1/p}.
>  \end{equation}
> By application of Theorem 1.1 in [Goldfel et al. Ann. Proba., 2024], a limit distribution theory for the one-sample reads as
> \begin{align} \sqrt{n}W\_p\big(\widehat{(\mathcal{R}\_{\boldsymbol{\rm u}} \mu)}\_n * \mathcal{N}\_{\sigma},\mathcal{R}\_{\boldsymbol{\rm u}} \mu* \mathcal{N}\_{\sigma}\big) \xrightarrow{\text{Law}} \sup\_{\varphi\in C\_0^\infty : \|{\varphi\|}\_{H^{1,q}(\mathcal{R}\_{\boldsymbol{\rm u}} \mu* \mathcal{N}\_{\sigma})} \leq 1} G\_{\mathcal{R}\_{\boldsymbol{\rm u}} \mu}(\varphi)
> \end{align}
> where $G\_{\mathcal{R}\_{\boldsymbol{\rm u}} \mu}(\varphi) = \{G\_{\mathcal{R}\_{\boldsymbol{\rm u}} \mu}(\varphi)\}\_{\varphi \in C\_0^\infty}$ is a centered Gaussian process whose paths are linear and continuous with respect to  the Sobolev seminorm $\|{\varphi\|}\_{H^{1,q}(\mathcal{R}\_{\boldsymbol{\rm u}}\mu* \mathcal{N}\_{\sigma})} =  {\|\nabla \varphi \|}\_{L\_q(\mathcal{R}\_{\boldsymbol{\rm u}}\mu* \mathcal{N}\_{\sigma})}$ and $q$ satisfies $1/p + 1/q = 1.$
> The latter result can't be applied to $W\_p(\hat{\hat\mu}\_{n}, \mathcal{R}\_{\boldsymbol{\rm u}} \mu * \mathcal{N}\_{\sigma\_k})$ that defines the one sample double empirical smoothed Gaussian sliced Wasserstein since the empirical distribution $\widehat{(\mathcal{R}\_{\boldsymbol{\rm u}} \mu)}\_n \neq \hat{\hat\mu}\_{n}$. We think that deriving limit distribution for $p$-smoothed Gaussian sliced requires an extensive theoretical investigations, that can be a future line direction of this work. We thank again the Reviewer for mentioning this point.
>
> **In [Nietert et al., ICML 2021], they show the convergence of transport plans. Can you derive similar results?**
>
> The results of Nietert et al. applies to the Gaussian Smoothed Wassertein distance. In our case, the smoothing is applied to the sliced Wasserstein distance. For latter distance, because of the expectation over the projection, there is no definition of a transport plan between the high-dimensional samples. Hence, we can not derive results on convergence of transport plans.
>
> **I suggest rewriting Proposition 3.16 as a stability result of $G\_{\sigma}SW^p$ as that in [Nietert et al., ICML 2021].**
>
> Our definition of $\text{G}\_{\sigma}\text{SW}^p$ involves a $p$-th power of the Wasserstein inside the integral over the unit sphere $\mathbb{S}^{d-1}$, that we can't obtain a similar result of stability as in [Nietert et al., ICML 2021]. Indeed, the stability result in [Nietert et al., ICML 2021] relies on a Minkoswki's inequality, $\big(\mathbf{E}[|U+V + W|^p]\big)^{\frac 1p} \geq \big(\mathbf{E}[|U+V|^p]\big)^{\frac 1p} - \big(\mathbf{E}[|W|^p]\big)^{\frac 1p}$, for any random variables $U, V, W \in \mathbb{L}\_p$ integrable. This couldn't be applied in our case, more specifically, if $U+V = (X\_{\boldsymbol{\rm u}}+ Z\_X) -(Y\_{\boldsymbol{\rm u}}+ Z\_Y)$ and $W=(Z'\_X -Z'\_Y)$, we will be blocked in the following stage:
> \begin{equation} \text{W}\_p^p(\mathcal{R}\_{\boldsymbol{\textrm{u}}} \mu * \mathcal{N}\_{\sigma\_2}, \mathcal{R}\_{\boldsymbol{\textrm{u}}} \nu * \mathcal{N}\_{\sigma\_2}) \geq
> \inf\_{\substack{X\_{\boldsymbol{\rm{u}}}, Z\_X, Z'\_X,\\
> Y\_{\boldsymbol{\textrm{u}}}, Z\_Y, Z'\_Y}} \Big(
> \Big(\mathbf{E}\big[\big|(X\_{\boldsymbol{\textrm{u}}} + Z\_X)  - (Y\_{\boldsymbol{\textrm{u}}} + Z\_Y) \big|^p\big]\Big)^{\frac 1p} - \Big(\mathbf{E}\big[|Z'\_X -Z'\_Y|^p\big]\Big)^{\frac{1}{p}} \Big)^p
> \end{equation}
> However, we use a triangle inequality fact stating that $\mathbf{E}[|U+V|^p] - 2^{p-1}\mathbf{E}[|W|^p]\leq 2^{p-1}\mathbf{E}[|U+V+W|^p]$.
> We rewrite  Proposition 3.16 as follows, and in the particular $p=1$ we get a proper stability result.
> > Let $0\leq \sigma\_1\leq \sigma\_2$ be two noise levels. Then, one has $\text{G}\_{\sigma\_2}\text{SW}\_p(\mu,\nu) \leq \text{G}\_{\sigma\_1}\text{SW}\_p(\mu,\nu)$ and
> \begin{align}
> |\text{G}\_{\sigma\_1}\text{SW}\_p(\mu,\nu) - \text{G}\_{\sigma\_2}\text{SW}\_p(\mu,\nu)| &\leq (2^{1-\frac{1}{p}}-1) \text{G}\_{\sigma\_2}\text{SW}\_p(\mu,\nu)+ 2^{\frac{5}{2}} (\sigma\_2^2 - \sigma\_1^2)^{},
> \end{align}
> in particular for $p=1$, $|\text{G}\_{\sigma\_1}\text{SW}(\mu,\nu) - \text{G}\_{\sigma\_2}\text{SW}(\mu,\nu)| \leq 2^{\frac{5}{2}} (\sigma\_2^2 - \sigma\_1^2)^{}.$

---

### Review · Reviewer_btsZ · 2024-08-14

**Summary Of Contributions:**

This works studies the theoretical properties of the Gaussian smoothed sliced Wasserstein distance as well as generalized versions, which are known as Gaussian-smoothed sliced divergences. The work provides a theoretical analysis of general Gaussian-smoothed sliced divergences. The analysis consists of

1) Establishing the topological properties of general Gaussian-smoothed sliced divergences
2) Proving sample complexity results of these divergences by introducing a double empirical distribution
3) Proving that these divergences satisfy an order relation with respect to a noise level and are also continuous with respect to this parameter.

The novelty of this work is that it studies the theoretical properties of Gaussian-smoothed sliced divergences. Previous work has focused on the Wasserstein distance, sliced Wasserstein distance, and Gaussian smoothed Wasserstein distance, but NOT Gaussian-smoothed sliced Wasserstein distance. This work aims to fill in this gap.

**Audience:**

Yes

**Broader Impact Concerns:**

No concerns

**Claims And Evidence:**

Yes

**Requested Changes:**

1) If the authors could try to clean up notation, that would be appreciated
2) Reword the introduction to make it clear that the results are general and not for just Gaussian smoothed sliced Wasserstein distances. It just happens to be that since the work applies to many Gaussian smoothed sliced divergences they are able to prove results for the popular Wasserstein distance.
3)

**Strengths And Weaknesses:**

**Strengths**

1. For someone not an expert in this area, I really enjoyed reading this paper. It was very clear from the start and the authors did a great job at making it clear where the contribution of this paper lies with respect to other work.

2. Many of the theorems and propositions are clearly explained.

3. For a mainly theoretical contribution, I appreciate that the authors also did some experimental analysis.

**Weaknesss**

1. The notation can get a bit overwhelming. I assume that it is almost necessary but if the authors in any way could make it a bit more easy to understand it would improve the clarity of the work.

2. The introduction says "In this work, we fill this gap by providing a theoretical analysis of the Gaussian smoothed sliced Wasserstein distance." However in actuality, the authors provide a theoretical analysis of Gaussian smoothed sliced divergences, which are more general. For that reason, I would think of removing this sentence or changing it because the authors provide results that are even more general and not specifically related to just Gaussian smoothed sliced Wasserstein distances! It almost felt like the authors were selling themselves short because the theory was a lot more general and did not just apply to the Wasserstein distance.

3. In section 3.2, the authors quantify

1. the convergence of the double empirical to GσSD(μ, ν)
2. (ii) the convergence of \hat{GσSD(μ, ν)}  to GσSD(μ, ν),when approximating the expectation over the random projection with sample mean.

Could you explain what the difference is between the two? It remains a little unclear which of them (if not both) is the main useful result in practice (if there is one).

---

> ### Author Response · Authors · 2024-09-03
>
> We thank the Reviewer for his time, consideration, and invaluable feedback. We provide point-by-point answers to the concerns pointed.
>
> **The notation can get a bit overwhelming. I assume that it is almost necessary but if the authors in any way could make it a bit more easy to understand it would improve the clarity of the work.**
>
> We agree that the notations are a bit heavy. However, we believe that we don't have any superfluous notations as the smoothing operation and the double empirical processes call for such notations (the convolution and the double hats...).
>
> **Reword the introduction to make it clear that the results are general and not for just Gaussian smoothed sliced Wasserstein distances. It just happens to be that since the work applies to many Gaussian smoothed sliced divergences they are able to prove results for the popular Wasserstein distance.**
>
> We have tried to reword the paper and provided more explanations/intuitions in order to provide some clarifications.
>
> **The introduction says "In this work, we fill this gap by providing a theoretical analysis of the Gaussian smoothed sliced Wasserstein distance." However, in actuality, the authors provide a theoretical analysis of Gaussian smoothed sliced divergences, which are more general. For that reason, I would think of removing this sentence or changing it because the authors provide results that are even more general and not specifically related to just Gaussian smoothed sliced Wasserstein distances! It almost felt like the authors were selling themselves short because the theory was a lot more general and did not just apply to the Wasserstein distance**
>
> Thanks for pointing this out. We have reworded the introduction and proposed a more general statement when possible.
>
> **In section 3.2, the authors quantify (i) the convergence of the double empirical to GσSD(μ, ν) (ii) the convergence of \hat{GσSD(μ, ν)} to GσSD(μ, ν), when approximating the expectation over the random projection with the sample mean.
> Could you explain what the difference is between the two? It remains a little unclear which of them (if not both) is the main useful result in practice (if there is one).**
>
> The first quantity aims at analyzing the sample complexity of the Gaussian smoothed Sliced Divergence whereas the second one aims at analyzing the approximation error due to the Monte-Carlo scheme considered for approximating the expectation.
> To make it more clearer, we added in the paper the following paragraph:
>
> > Given the results of Propositions 3.11 and 3.12, we provide a finer analysis of $\text{G}\_{\sigma}\text{SW}\_p(\mu,\nu)$'s sample complexity. Towards this end, for a fixed random projection $\boldsymbol{\textrm{u}}\_l, (1\leq l \leq L)$ we define $\hat{\hat{\mu}}\_{n,l} = \frac 1n \sum\_{i=1}^n \delta\_{\boldsymbol{\rm{u}}\_l^\top X_i + Z^x_i}$ (similarly for $\hat{\hat\nu}\_{n,l})$  and set
> \begin{equation}
> \widehat{\hat{\text{G}}\_{\sigma}\text{SW}_p}
>  (\hat \mu\_n,\hat\mu\_n) = \Big(
> \frac{1}{L}
> \sum\_{l=1}^L \text{W}^p_p(\hat{\hat\mu}\_{n,l}, \hat{\hat\nu}\_{n,l})
> \Big)^{1/p}.
> \end{equation}
> The overall complexity of $\text{G}\_{\sigma}\text{SW}\_p(\mu,\nu)$ consists in its approximation by sampling and projection of the origin probability measures $\mu, \nu$, i.e. through $\widehat{\hat{\text{G}}\_{\sigma}\text{SW}\_p}
>  (\hat \mu\_n,\hat\mu\_n).$  By application of triangle inequality, one has
> \begin{equation}
> |\widehat{\hat{\text{G}}\_{\sigma}\text{SW}\_p}^p
>  (\hat \mu\_n,\hat\mu\_n) - \text{G}\_{\sigma}\text{SW}\_p^p(\mu,\nu)|\\
>  \leq |\widehat{\hat{\text{G}}\_{\sigma}\text{SW}\_p}^p
>  (\hat \mu\_n,\hat\mu\_n) - \hat{\text{G}}\_{\sigma}\text{SW}^p\_p
>  (\hat \mu\_n,\hat\mu\_n)| +  |\hat{\text{G}}\_{\sigma}\text{SW}\_p^p
>  (\hat \mu\_n,\hat\mu\_n) - \text{G}\_{\sigma}\text{SW}^p\_p(\mu,\nu)|
> \end{equation}
> The first term in the right-hand-side (RHS) of the latter decomposition can be controlled by Proposition 3.12, in the following way:
> \begin{equation}
> \mathbf{E}\_{u\_d^{\otimes\_L}}\big[|\widehat{\hat{\text{G}}\_{\sigma}\text{SW}\_p}^p (\hat \mu\_n,\hat\mu\_n) - \hat{\text{G}}\_{\sigma}\text{SW}^p\_p(\hat \mu\_n,\hat\mu\_n)|\big]\leq \frac{\hat A(p,\sigma)}{\sqrt{L}},
> \end{equation}
> where $\hat A^2(p,\sigma) = \mathbf{V}\_{\boldsymbol{\rm u} \sim u\_d}[\text{W}\_p^p(\hat{\hat\mu}\_{n}, \hat{\hat\nu}\_{n})]$. However we don't have a proper control for $p\geq 2$ of the second term in the RHS, $|\hat{\text{G}}\_{\sigma}\text{SW}\_p^p(\hat \mu\_n,\hat\mu\_n) - \text{G}\_{\sigma}\text{SW}^p\_p(\mu,\nu)|$,  as it can be seen from Proposition 3.11. For that reason, we derive an overall complexity of order $O(n^{-1/2} + L^{-1/2})$ in the case of $p=1$.*
>
> As stated by **Reviewer cJ1H**, both terms are of importance and provide an intuition on how the empirical divergence behaves with the number of samples at disposal.
>
> Nonetheless, a practitioner can only play with the number of projections, and corollary 3.13 provides a good rule of thumbs for $L$.

---

### Review · Reviewer_cJ1H · 2024-08-20

**Summary Of Contributions:**

This paper presents theoretical analysis and some computational experiments on Gaussian-smoothed sliced probability divergences, which include Gaussian-smoothed Wasserstein distances as a special case. The theoretical results include general properties of the divergences and some approximation complexity results.

**Audience:**

Yes

**Claims And Evidence:**

Yes

**Requested Changes:**

Get more explicit about the total complexity bounds.

**Strengths And Weaknesses:**

# Strengths

- The results are well-motivated. I can see this being of interest to many people in the field.
- The results appear to be rigorous
- The writing is fairly clear and well-organized
- The experiments support the theory

# Weaknesses

- In my view, the most interesting approximation result is the total complexity bound from Corollary 3.13. The paper would be stronger if this were stated with explicit constants and proved explicitly.
- The proofs appear to be rigorous, but are quite terse and dense. A bit more discussion might help.

---

> ### Author Response · Authors · 2024-09-03
>
> We thank the Reviewer for his time, consideration, and invaluable feedback. Next, we provide point-by-point answers to the concerns pointed out by the Reviewer.
>
> **In my view, the most interesting approximation result is the total complexity bound from Corollary 3.13. The paper would be stronger if this were stated with explicit constants and proved explicitly.**
>
> Thanks for this comment. Please see the details given in the answers to **Reviewer btsZ**. We have provided the proof of this corollary when $p=1$ in appendix A.6 where we explicit the constants of the order $O(n^{-1/2} + L^{-1/2})$.
>
>
> **The proofs appear to be rigorous but are quite terse and dense. A bit more discussion might help.**
>
> Thanks for pointing this out.  To make some proofs more readable, we added an overall structure and explicit the technical material to be used.

---

### Author Response · Authors · 2024-09-03

We want to thank all the reviewers for the constructive feedback you sent us. In this revised version, we considered and addressed all the comments, and we believe they helped us greatly improve the paper. In what follows we provide detailed answers to all the questions raised in the referees' reports. First, let us highlight the main changes we made to the manuscript:
- We ameliorate the Introduction and highlight that the results are for general smoothed Gaussian sliced measures.
 - In all the definitions of the distances/divergences, we remove the superscript $p$ that refers to a power and make as a subscript to denote the $p$-th order, for example the notation $\text{G}\_{\sigma}\text{SW}^p$ becomes ${\text{G}}\_{\sigma}\text{SW}\_p$.
- We consequently adjust the bounds in the propositions with respect to $p$-th order of the considered distances/divergences.
- We detail the overall complexity for the Gaussian smoothed sliced Wasserstein as proposed by **Reviewer btsZ** and **Reviewer cJ1H**. We also give a proof of Corollary 3.13 in case $p=1.$
- We rewrite Proposition 3.16 as a stability result as suggested by **Reviewer vY8A**.
 - To make some proofs more readable, we added an overall structure and explicit the technical material to be used.

---

### Decision · Action_Editor_9MMo · 2024-10-21

**Recommendation:** Accept as is

**Comment:**

Thank you for your submission!  The reviewers all felt that this would make a solid addition to TMLR.

**Audience:**

The reviewers agreed that the paper is well in scope for TMLR.

**Claims And Evidence:**

Reviewers agreed that claims were accurate and convincing.